# NeuroLKH: Combining Deep Learning Model with Lin-Kernighan-Helsgaun Heuristic for Solving the Traveling Salesman Problem

**Liang Xin**
Nanyang Technological University
Singapore
XINL0003@e.ntu.edu.sg

**Wen Song**
Shandong Unviersity
Qingdao, China
wensong@email.sdu.edu.cn

**Zhiguang Cao**[*]
Singapore Institute of Manufacturing Technology
Singapore
zhiguangcao@outlook.com

**Jie Zhang**
Nanyang Technological University
Singapore
jzhang@ntu.edu.sg

## Abstract

We present NeuroLKH, a novel algorithm that combines deep learning with the strong traditional heuristic Lin-Kernighan-Helsgaun (LKH) for solving Traveling Salesman Problem. Specifically, we train a Sparse Graph Network (SGN) with supervised learning for edge scores and unsupervised learning for node penalties, both of which are critical for improving the performance of LKH. Based on the output of SGN, NeuroLKH creates the edge candidate set and transforms edge distances to guide the searching process of LKH. Extensive experiments firmly demonstrate that, by training one model on a wide range of problem sizes, NeuroLKH significantly outperforms LKH and generalizes well to much larger sizes. Also, we show that NeuroLKH can be applied to other routing problems such as Capacitated Vehicle Routing Problem (CVRP), Pickup and Delivery Problem (PDP), and CVRP with Time Windows (CVRPTW).

## 1 Introduction

Traveling Salesman Problem (TSP) is an important NP-hard Combinatorial Optimization Problem with extensive industrial applications in various domains. Exact methods have the exponential worst-case computational complexity, which renders them impractical for solving large-scale problems in reality, even for highly optimized solvers such as Concorde. In contrast, although lacking optimality guarantees and non-trivial theoretical analysis, heuristic solvers search for near-optimal solutions with much lower complexity. They are usually desirable for real-life applications where statistically better performance is the goal.

Traditional heuristic methods are manually designed based on expert knowledge which is usually human-interpretable. However, supported by the recent development of deep learning technology, modern methods train powerful deep neural networks to learn the complex patterns from the TSP instances generated from some specific distributions [32, 1, 6, 21, 18, 34, 33, 35]. The performances of deep learning models for solving TSP are constantly improved by these works, which unfortunately are still far worse than the strong traditional heuristic solver and generally limited to relatively small problem sizes.

---

[*]Zhiguang Cao is the corresponding author.

35th Conference on Neural Information Processing Systems (NeurIPS 2021).

We believe that learning-based methods should be combined with strong traditional heuristic algorithms, which is also suggested by [2]. In such a way, while learning the complex patterns from data samples, the efficient heuristics highly optimized by researchers for decades can be effectively utilized, especially for problems such as TSP which are well-studied due to their importance.

The Lin-Kernighan-Helsgaun (LKH) algorithm [12, 13] is generally considered as a very strong heuristic for solving TSP, which is developed based on the Lin-Kernighan (LK) heuristic [25]. LKH iteratively searches for $\lambda$-opt moves to improve the existing solution where $\lambda$ edges of the tour are exchanged for another $\lambda$ edges to form a shorter tour. To save the searching time, the edges to add are limited to a small *edge candidate set*, which is created before search. One of the most significant contributions of LKH is to generate the edge candidate set based on Minimum Spanning Tree, rather than using the nearest neighbor method in the LK heuristic. Furthermore, LKH applies penalty values to the nodes which are iteratively optimized using subgradient optimization (will be detailed in Section 3). The optimized node penalties are used by LKH to transform the edge distances for the $\lambda$-opt searching process and improve the quality of edge candidate sets, both of which help find better solutions.

However, the edge candidate set generation in LKH is still guided by hand-crafted rules, which could limit the quality of edge candidates and hence the search performance. Moreover, the iterative optimization of node penalties is time-consuming, especially for large-scale problems. To address these limitations, we propose NeuroLKH, a novel learning-based method featuring a Sparse Graph Network (SGN) combined with the highly efficient $\lambda$-opt local search of LKH. SGN outputs the edge scores and node penalties simultaneously, which are trained by supervised learning and unsupervised learning, respectively. NeuroLKH transforms the edge distances based on the node penalties learned inductively from training instances, instead of performing iterative optimization for each instance, therefore saving a significant amount of time. More importantly, at the same time the edge scores are used to create the edge candidate set, leading to substantially better sets than those created by LKH. NeuroLKH trains one single network on TSP instances across a wide range of sizes and generalizes well to substantially larger problems with minutes of unsupervised offline fine-tuning to adjust the node penalty scales for different sizes.

Same as existing works on deep learning models for solving TSP, NeuroLKH aims to learn complex patterns from data samples to find better solutions for instances following specific distributions. Following the evaluation process in these works, we perform extensive experiments. Results show that NeuroLKH improves the baseline algorithms by large margins, not only across the wide range of training problem sizes, but also on much larger problem sizes not used in training. Furthermore, NeuroLKH trained with instances of relatively simple distributions generalizes well to traditional benchmark with various node distributions such as the TSPLIB [27]. Also, we show that NeuroLKH can be applied to guide the extension of LKH [14] for more complicated routing problems such as the Capacitated Vehicle Routing Problem (CVRP), Pickup and Delivery Problem (PDP) and CVRP with Time Windows (CVRPTW), using generated test datasets and traditional benchmarks [28, 30].

## 2   Related works

Till now, for routing problems such as TSP, most works focus on learning construction heuristics, where deep neural networks are trained to sequentially select the nodes to visit with supervised learning [32, 15] or reinforcement learning [1, 6, 26, 21, 22]. Similarly, networks are trained to pick edges in [18, 20]. In another line of works [4, 33, 16, 9, 5], researchers employ deep learning models to learn the actions for improving existing solutions, such as picking regions and rules or selecting nodes for the 2-opt heuristic. However, the performance of these works is still quite far from the strong non-learning heuristics such as LKH. In addition, they focus only on relatively small-sized problems (up to hundreds of nodes).

A recent work [8] generalizes a network pre-trained on fixed-size small graphs to solve larger size problems by sampling small sub-graphs to infer and merging the results. This interesting idea can be applied to very large graphs, however, the performance is still inferior to LKH and deteriorates rapidly with the increase of problem size.

In a concurrent work [36], a VSR-LKH method is proposed which also applies a learning method in combination with LKH. However, very different from our method, VSR-LKH applies traditional reinforcement learning during the searching process *for each instance*, instead of learning patterns for

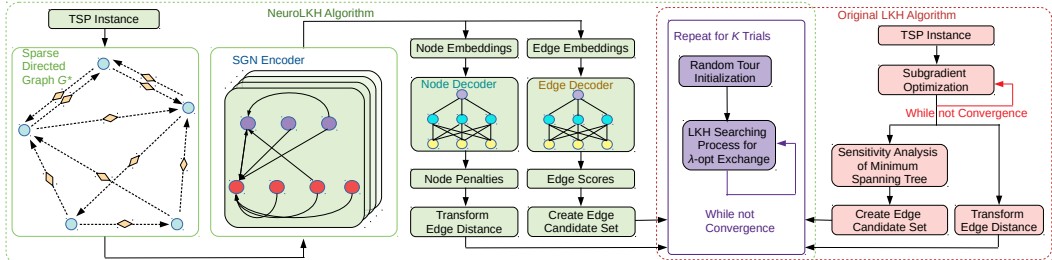

Figure 1: NeuroLKH algorithm and the original LKH algorithm.

a class of instances. Moreover, VSR-LKH aims to guide the decision on edge selections within the edge candidate set, which is generated using the original procedure of LKH. NeuroLKH significantly outperforms VSR-LKH by large margins in all the settings of our experiments on testing instances following the training distributions, especially when the time limits are short. Even more impressively, NeuroLKH achieves performance similar to VSR-LKH on traditional benchmark TSPLIB [27] with various node distributions, which are very different from the training distributions for NeuroLKH.

## 3   Preliminaries: LKH algorithm

The Lin-Kernighan-Helsgaun (LKH) algorithm [12, 13] is a local optimization algorithm developed based on the $\lambda$-opt move [24], where $\lambda$ edges in the current tour are exchanged by another set of $\lambda$ edges to achieve a shorter tour. While solving one instance, the LKH algorithm can conduct multiple trials to find better solutions. In each trial, starting from a randomly initialized tour, it iteratively searches for $\lambda$-opt exchanges that improve the tour, until no such exchanges can be found. In each iteration, the $\lambda$-opt exchanges are searched in the ascending order of variable $\lambda$ and the tour will be replaced once an exchange is found to reduce the tour distance.

One central rule is that the $\lambda$-opt searching process is restricted and directed by an edge candidate set, which is created before search based on the $\alpha$-measure using sensitivity analysis of the Minimum Spanning Tree. Here we briefly introduce the related concepts. A TSP graph can be viewed as an undirected graph $G = (V, E)$ with $V$ as the set of $|V|$ nodes and $E$ as the set of edges weighted by distances. A spanning tree of $G$ is a connected graph with $|V| - 1$ edges from $G$ and no cycles where any pair of nodes is connected by a path. A 1-tree of $G$ is a spanning tree for the graph of node set $V \backslash \{1\}$ combined with two edges in $E$ connected to node 1, an arbitrary special node in $V$. A minimum 1-tree is the 1-tree with minimum length. The $\alpha$-measure of an edge $(i, j) \in E$ for graph $G$ is defined as $\alpha(i, j) = \mathbb{L}(T^+(i, j)) - \mathbb{L}(T)$, where $\mathbb{L}(T)$ is the length of Minimum 1-Tree $T$ and $\mathbb{L}(T^+(i, j))$ is the length of Minimum 1-Tree $T^+(i, j)$ required to include the edge $(i, j)$. The $\alpha$-measure of an edge can be viewed as the extra length of the Minimum 1-Tree to include this edge.

The edge candidate set consists of the $k$ edges with the smallest $\alpha$-measures connected to each node ($k = 5$ as default). During the $\lambda$-opt searching process, the edges to be included into the new tour are limited to the edges in this candidate set, and edges with smaller $\alpha$-measures will have higher priorities to be searched over. Therefore this candidate set not only restricts but also directs the search.

Moreover, the quality of $\alpha$-measures can be improved significantly by a subgradient optimization method. If we add a penalty $\pi_i$ to each node $i$ and transform the original distance $s_{i,j}$ of the edge $(i, j)$ to a new distance $c_{i,j}$ as $c_{i,j} = s_{i,j} + \pi_i + \pi_j$, the optimal tour for the TSP will stay the same but the Minimum 1-Tree usually will change. Because by definition, a Minimum 1-Tree with node degrees all equal to 2 is an optimal solution for the corresponding TSP instance. With the length of Minimum 1-Tree resulting from the penalty $\pi = (\pi_1, ..., \pi_{|V|})$ as $\mathbb{L}(T_\pi)$, $w(\pi) = \mathbb{L}(T_\pi) - 2\Sigma_i \pi_i$ is a lower bound of the optimal tour distance for the original TSP instance. LKH applies subgradient optimization [11] to iteratively maximize this lower bound for multiple steps until convergence by applying $\pi^{\tau+1} = \pi^\tau + t^\tau(d^\tau - 2)$ at step $\tau$, where $t^\tau$ is the scalar step size, $d^\tau$ is the vector of node degrees in the Minimum 1-Tree with penalty $\pi^\tau$. Therefore, the node degrees are pushed towards 2. The $\alpha$-measures after this optimization will substantially improve the quality of edge candidate set. Furthermore, the transformed edge distance $c_{i,j}$ after this optimization helps find better solutions when used during the searching process for $\lambda$-opt exchanges.

# 4 The proposed NeuroLKH algorithm

The subgradient optimization in LKH can substantially improve the quality of edge candidate sets based on the $\alpha$-measures, and transform the edge distances effectively to achieve reasonably good performance. However, it still has major limitations as the optimization process is over one instance iteratively until convergence, which costs a large amount of time, especially for large-scale problems. Moreover, even after subgradient optimization, some critical patterns could be missed by the relatively straightforward sensitivity analysis of spanning tree. Therefore, the quality of edge candidate set could be further improved by large margins, which will in turn improve the overall performance.

We propose the NeuroLKH algorithm, which employs a Sparse Graph Network to learn the complex patterns associated with the TSP instances generated from a distribution. Concretely, the network will learn the edge scores and node penalties simultaneously with a multi-task training process. The edge scores are trained with supervised learning for creating the edge candidate set, while the node penalties are trained with unsupervised learning for transforming the edge distances. The architecture of NeuroLKH is presented in Figure 1, along with the original LKH algorithm. We will detail the Sparse Graph Network, the training process and the proposed NeuroLKH algorithm in the following.

## 4.1 Sparse Graph Network

For the Sparse Graph Network (SGN), we format the TSP instance as a sparse directed graph $G^* = (V, E^*)$ containing the node set $V$ and a sparse edge set $E^*$ which only includes the $\gamma$ shortest edges pointed from each node, as shown in the leftmost green box in Figure 1, where the circles represent the nodes and the diamonds represent the directed edges. Sparsification of the graph is crucial for effectively training the deep learning model on large TSP instances and generalizing to even larger sizes. Note that edge $(i, j)$ belongs to $E^*$ does not necessarily mean that the opposite-direction edge $(j, i)$ belongs to $E^*$. The node inputs $x_v \in \mathbb{R}^2$ are the node coordinates and the edge inputs $x_e \in \mathbb{R}$ are the edge distances. Though we focus on 2-dimensional TSP with Euclidean distance as the other deep learning literature like [21], the model can be applied to other kinds of TSP.

The SGN consists of 1) one encoder embedding the edge and node inputs into the corresponding feature vectors, and 2) two decoders for the edge scores and node penalties, respectively.

**Encoder.** The encoder first linearly projects the node inputs $x_v$ and the edge inputs $x_e$ into feature vectors $v_i^0 \in \mathbb{R}^D$ and $e_{i,j}^0 \in \mathbb{R}^D$, respectively, where $D$ is the feature dimension, $i \in V$ and $(i, j) \in E^*$. Then the node and edge features are embedded with $L$ Sparse Graph Convolutional Layers, which are defined formally as follows:

$$attn_{i,j}^l = exp(W_a^l e_{i,j}^{l-1}) \oslash \sum_{(i,m) \in E*} exp(W_a^l e_{i,m}^{l-1}), \tag{1}$$

$$v_i^l = v_i^{l-1} + ReLU(BN(W_s^l v_i^{l-1} + \sum_{(i,j) \in E^*} attn_{i,j}^l \odot W_n^l v_j^{l-1})), \tag{2}$$

$$r_{i,j}^l = \begin{cases} W_r^l e_{j,i}^{l-1}, & \text{if } (j,i) \in E^* \\ W_r^l p^l, & \text{otherwise} \end{cases} \tag{3}$$

$$e_{i,j}^l = e_{i,j}^{l-1} + ReLU(BN(W_f^l v_i^{l-1} + W_t^l v_j^{l-1} + W_o^l e_{i,j}^{l-1} + r_{i,j}^l)), \tag{4}$$

where $\odot$ and $\oslash$ represent the element-wise multiplication and the element-wise division, respectively; $l = 1, 2, ..., L$ is the layer index; $W_a^l, W_s^l, W_n^l, W_r^l, W_f^l, W_t^l, W_o^l \in \mathbb{R}^{D \times D}$ and $p^l \in \mathbb{R}^D$ are trainable parameters; Eqs. (2) and (4) consist of a Skip-Connection layer [10] and a Batch Normalization layer [17] in each; and the idea of element-wise attention in Eq. (1) is adopted from [3]. As the input graph $G^*$ is directed and sparse, edges with different directions are embedded separately. But obviously the embedding of an edge $(i, j)$ should benefit from knowing whether its opposite-direction counterpart $(j, i)$ is also in the graph and the information of $(j, i)$, which motivates our design of Eqs. (3) and (4).

**Decoders.** The edge decoder takes the edge embeddings $e_{i,j}^L$ from the encoder and embeds them with two layers of linear projection followed by ReLU activation into $e_{i,j}^f$. Then the edge scores $\beta_{i,j}$ are calculated as follows:

$$\beta_{i,j} = \frac{exp(W_\beta e_{i,j}^f)}{\sum_{(i,m) \in E^*} exp(W_\beta e_{i,m}^f)}, \tag{5}$$

---

**Algorithm 1** NeuroLKH Algorithm

---

**Input**: TSP instance, number of trials $K$
**Output**: TSP solution $BestTour$

1: Convert the TSP instance to SGN input $G^*, x_v, x_e$
2: Calculate the edge scores $\beta_{i,j}$ and the node penalties $\pi_i$ with Eqs. (5) and (6)
3: $EdgeDistance$=TransformEdgeDistance($\pi$)
4: $EdgeCandidateSet$=CreateEdgeCandidateSet($\beta$)
5: $BestTour$=LKHSearchingTrials($EdgeDistance, EdgeCandidateSet, K$)
6: **return** $BestTour$

---

Similarly, the node decoder first embeds the node embeddings $v_i^L$ with two layers of linear projection and ReLU activation into $v_i^f$. Then the node penalties $\pi_i$ are calculated as follows:

$$\pi_i = C \tanh(W_\pi v_i^f), \tag{6}$$

where $W_\beta, W_\pi \in \mathbb{R}^{D \times 1}$ are trainable parameters; $C = 10$ is used to keep the node penalties in the range of $[-10, 10]$.

## 4.2 Training process

We train the network to learn the edge scores with supervised learning. And the edge loss $\mathcal{L}_\beta$ is detailed as follows:

$$\mathcal{L}_\beta = -\frac{1}{\gamma |V|} \sum_{(i,j) \in E^*} (\mathbb{1}\{(i,j) \in E_o^*\} \log \beta_{i,j} + \mathbb{1}\{(i,j) \notin E_o^*\} \log(1 - \beta_{i,j})), \tag{7}$$

where $E_o^* = \{(i,j) \in E^* | (i,j) \text{ in the optimal tour}\}$. Effectively, we increase the edge scores $\beta_{i,j}$ if the edge $(i,j)$ belongs to the optimal tour and decrease them otherwise.

The node penalties are trained by unsupervised learning. Similar to the goal of subgradient optimization in LKH, we are trying to transform the Minimum 1-Tree generated from the TSP graph $G$ closer to a tour where all nodes have a degree of 2. An important distinction from LKH is that we are learning the patterns for a class of TSP instances following a distribution, instead of optimizing the penalties for a specific TSP instance. The node loss $\mathcal{L}_\pi$ is detailed as follows:

$$\mathcal{L}_\pi = -\frac{1}{|V|} \sum_{i \in V} (d_i(\pi) - 2)\pi_i, \tag{8}$$

where $d_i(\pi)$ is the degree of node $i$ in the Minimum 1-Tree $T_\pi$ induced with penalty $\pi = (\pi_1, ..., \pi_{|V|})$. The penalties are increased for nodes with degrees larger than 2 and decreased for nodes with smaller degrees. The SGN is trained for the task of outputting the edge scores and node penalties simultaneously with the loss function $\mathcal{L} = \mathcal{L}_\beta + \eta_\pi \mathcal{L}_\pi$, where $\eta_\pi$ is the coefficient for balancing the two losses.

## 4.3 NeuroLKH algorithm

The process of using NeuroLKH to solve one instance is shown in Algorithm 1. Firstly, the TSP instance is converted to a sparse directed graph $G^*$. Then the SGN encoder embeds the nodes and edges in $G^*$ into feature embeddings, based on which the decoders output the node penalties $\pi$ and edge scores $\beta$. Afterwards, NeuroLKH creates powerful edge candidate set and transforms the distance of each edge effectively, which further guides NeuroLKH to conduct multiple LKH trials to find good solutions. We detail each part as follows.

**Transform Edge Distance.** Based on the node penalties $\pi_i$, the original edge distances $s_{i,j}$ are transformed into new distances $c_{i,j} = s_{i,j} + \pi_i + \pi_j$, which will be used in the search process. With such a transformation, the optimal solution tour will stay the same. And the tour distance calculated with the transformed edge distances will be subtracted by $2\sum_{i \in V} \pi_i$ to restore the tour distance for the original TSP.

**Create Edge Candidate Set.** For each node $i \in V$, the edge scores $\beta_{i,j}$ are sorted for $(i,j) \in E^*$ and the edges with the top-$k$ largest scores are included in the edge candidate set. Edges with larger

scores have higher priorities in the candidate set, which will be tried first for adding in the exchange during the LKH search process. Note that neither the original LKH nor NeuroLKH can guarantee all the edges in the optimal tour to be included in the edge candidate set. However, optimal solutions are still likely to be found during the multiple trials.

**LKH Searching Trials.** To solve one TSP instance, LKH conducts multiple trials to find better solutions. In each trial, one tour is initialized randomly, and iterations of LKH search are conducted for the $\lambda$-opt exchanges until the tour can no longer be improved by such exchanges. In each iteration, LKH searches in the ascending order of $\lambda$ for $\lambda$-opt exchanges to reduce tour length, which will be applied once found.

Based on the trained SGN network, NeuroLKH infers the edge distance transformation and candidate set to guide the LKH trials, which is done by performing forward calculation through the model. This is much faster than the corresponding procedure in the original LKH, which employs subgradient optimization on each instance iteratively until convergence and is apparently time-consuming especially for large-scale problems. More importantly, rather than using the hand-crafted rules based on sensitivity analysis in the original LKH, NeuroLKH learns to create edge candidate set of much higher quality with the powerful deep model, leading to significantly better performance.

# 5    Experiments

In this section, we conduct extensive experiments on TSP with various sizes and show the effective performance of NeuroLKH compared to the baseline algorithms. Our code is publicly available.[1]

**Dataset distribution.** Closely following the existing works such as [21], we experiment with the 2-dimensional TSP instances in the Euclidean distance space where both coordinates of each node are generated independently from a unit uniform distribution. We train only one network using TSP instances ranging from 101 to 500 nodes. Since the amount of supervision and feedback during training is linearly related to the number of nodes. We generate $500000/|V|$ instances for each size $|V|$ in the training dataset, resulting in approximately 780000 instances in total. Therefore the amounts of supervision and feedback are kept similar across different sizes. We use Concorde [2] to get the optimal edges $E_o^*$ for the supervised training of edge scores. For testing, we generate 1000 instances for each testing problem size.

**Hyperparameters.** We choose the number of directed edges pointed from one node in the sparse edge set $E^*$ as $\gamma = 20$, which results in only 0.01% of the edges in the optimal tours missed in $E^*$ for the training dataset. We also conduct experiments to justify this choice in Appendix Section A. The hidden dimension is set to $D = 128$ in the network with $L = 30$ Sparse Graph Convolutional Layers. The node penalty coefficient in the loss function is $\eta_\pi = 1$. The network is trained by Adam Optimizer [19] with learning rate of 0.0001 for 16 epochs, which takes approximately 4 days. The deep learning models are trained and evaluated with one RTX-2080Ti GPU. The other parts of experiments without deep models for NeuroLKH and other baselines are conducted with random seed 1234 on an Intel(R) Core(TM) i9-10940X CPU unless stated otherwise. Hyperparameters for the LKH searching process are consistent with the example script for TSP given by LKH available online [3] and those used in [36].

## 5.1    Comparative study on TSP

Here, we compare NeuroLKH with the original LKH algorithm [13] and the recently proposed VSR-LKH algorithm [36]. We do not compare with other deep learning based methods here because their performances are rather inferior to LKH, and most of them can hardly generalize to problems with more than 100 nodes. One exception is the method in [8], which is tested on large problems but the performances are still far worse than LKH.

All algorithms are run once for each testing instance as we find running multiple times only provides very marginal improvement. For each testing problem size, we run the original LKH for 1, 10, 100, and 1000 trials, and record the total amounts of time in solving the 1000 instances. Then we impose

---

[1]https://github.com/liangxinedu/NeuroLKH
[2]https://www.math.uwaterloo.ca/tsp/concorde
[3]http://akira.ruc.dk/%7Ekeld/research/LKH-3/LKH-3.0.6.tgz

Table 1: Comparative results on training sizes

| Method | $\|V\|=100$ | | | $\|V\|=200$ | | | $\|V\|=500$ | | |
|---|---|---|---|---|---|---|---|---|---|
| | Time(s) | Obj | Gap(‰) | Time(s) | Obj | Gap(‰) | Time(s) | Obj | Gap(‰) |
| Concorde | 207 | *7.753246 | 0.000 | 1072 | *10.701303 | 0.000 | 17022 | *16.541830 | 0.000 |
| LKH (1 trial) | | 7.755071 | 2.353 | | 10.707043 | 5.364 | | 16.556733 | 9.009 |
| VSR-LKH | 33 | 7.754980 | 2.236 | 80 | 10.706739 | 5.080 | 338 | 16.557297 | 9.350 |
| NeuroLKH | | **7.753332** | **0.111** | | **10.701873** | **0.533** | | **16.543197** | **0.826** |
| LKH (10 trials) | | 7.754177 | 1.200 | | 10.703724 | 2.263 | | 16.548017 | 3.740 |
| VSR-LKH | 43 | 7.754184 | 1.209 | 111 | 10.703997 | 2.518 | 445 | 16.549591 | 4.692 |
| NeuroLKH | | **7.753311** | **0.083** | | **10.701623** | **0.299** | | **16.542880** | **0.634** |
| LKH (100 trials) | | 7.753450 | 0.263 | | 10.701755 | 0.423 | | 16.543707 | 1.134 |
| VSR-LKH | 127 | 7.753407 | 0.207 | 368 | 10.701687 | 0.359 | 1147 | 16.543085 | 0.759 |
| NeuroLKH | | **7.753270** | **0.030** | | **10.701381** | **0.073** | | **16.542163** | **0.201** |
| LKH (1000 trials) | | 7.753254 | 0.010 | | 10.701351 | 0.045 | | 16.542125 | 0.178 |
| VSR-LKH | 938 | 7.753322 | 0.097 | 2805 | 10.701336 | 0.031 | 7527 | 16.541934 | 0.063 |
| NeuroLKH | | **7.753247** | **0.000** | | **10.701303** | **0.000** | | **16.541847** | **0.010** |

the same amounts of time as time limits to NeuroLKH and VSR-LKH for solving the same 1000 instances for fair comparison. Note that for NeuroLKH, the solving time is the summation of the inference time of SGN on GPU and LKH searching time on CPU. In the following tables, for each size and time limit, we report the average performance (tour distance) and the total solving time for the 1000 testing instances.

**Comparison on training sizes.** In Table 1, we report the performances of LKH, VSR-LKH and NeuroLKH on three testing datasets with 100, 200 and 500 nodes, which are within the size range of instances used in training. Note that we train only one SGN Network on a wide range of problem sizes and here we use these three sizes to demonstrate the testing performances. We also use the exact solver Concorde on these instances to obtain the optimal solutions and compute the optimality gap for each method. As shown in this table, it is clear that NeuroLKH outperforms both LKH and VSR-LKH significantly and consistently across different problem sizes and with different time limits. Notably, the optimality gaps are reduced by at least an order of magnitude for most of the cases, which is a significant improvement.

**Generalization analysis on larger sizes.** We further show the generalization ability of NeuroLKH on much larger graph sizes of 1000, 2000 and 5000 nodes. Note that while the edge scores in SGN generalize well without any modification, it is hard for the node penalties to directly generalize. This is because they are trained unsupervisedly and SGN does not have any knowledge about how to penalize the nodes for larger TSP instances. Nevertheless, this could be resolved by a simple fine-tuning step. As the learned node embeddings are very powerful, we only fine-tune the very small amount of parameters in the SGN node decoder and keep the other parameters fixed. Specifically, for each of the large sizes, we fine-tune the node decoder for 100 iterations with batch size of $5000/|V|$, which only takes less than one minute for each size of 1000, 2000 and 5000. This fast fine-tuning process is for TSPs of one size generated from the distribution instead of specific instances, and may be viewed as adjusting the scale of penalties for large sizes. The generalization results are summarized in Table 2. Note that we do not run Concorde here due to the prohibitively long running time, and the gaps are with respect to the best value found by all methods. Clearly, NeuroLKH generalizes well to substantially larger problem sizes and the improvement of NeuroLKH over baselines is significant and consistent across all the settings.

**Further discussion.** The inference time of SGN in NeuroLKH for the 1000 instances of 100, 200, 500, 1000, 2000 and 5000 nodes is 3s, 6s, 16s, 33s, 63s and 208s, which is approximately linear with the number of nodes $|V|$. In contrast, the subgradient optimization in LKH and VSR-LKH needs 20s, 51s, 266s, 1028s, 4501s and 38970s, which grows superlinearly with $|V|$ and is much longer than SGN inference, especially for large-scale problems. For NeuroLKH, the saved time is used to conduct more trials, which effectively helps to find better solutions. This effect is more salient with short time limit. Meanwhile, the number of trials is also small for short time limit and the algorithm only searches a small number of solutions, in which case the guidance of edge candidate set is more important. Due to these two reasons, the improvement of NeuroLKH over baselines is particularly substantial for short time limit. This is a desirable property especially for time-critical applications and solving large-scale problems, for which large numbers of trials are not feasible.

Table 2: Comparative results on generalization sizes

| Method | Time(s) | $\|V\|=1000$ Obj | Gap(‰) | Time(s) | $\|V\|=2000$ Obj | Gap(‰) | Time(s) | $\|V\|=5000$ Obj | Gap(‰) |
|---|---|---|---|---|---|---|---|---|---|
| LKH (1 trial) | | 23.155916 | 10.593 | | 32.483851 | 11.264 | | 51.025519 | 12.284 |
| VSR-LKH | 1183 | 23.154946 | 10.173 | 4843 | 32.485551 | 11.788 | 40048 | 51.025539 | 12.288 |
| NeuroLKH | | **23.133494** | **0.899** | | **32.449752** | **0.755** | | **50.965382** | **0.484** |
| LKH (10 trials) | | 23.143435 | 5.197 | | 32.466953 | 6.056 | | 50.998721 | 7.026 |
| VSR-LKH | 1414 | 23.143347 | 5.159 | 5322 | 32.467997 | 6.377 | 41523 | 51.000093 | 7.295 |
| NeuroLKH | | **23.133066** | **0.714** | | **32.449519** | **0.683** | | **50.965219** | **0.452** |
| LKH (100 trials) | | 23.135427 | 1.735 | | 32.455454 | 2.512 | | 50.976677 | 2.700 |
| VSR-LKH | 2567 | 23.134426 | 1.302 | 7371 | 32.454427 | 2.195 | 47884 | 50.979317 | 3.218 |
| NeuroLKH | | **23.132258** | **0.365** | | **32.448666** | **0.420** | | **50.964677** | **0.345** |
| LKH (1000 trials) | | 23.132216 | 0.347 | | 32.448954 | 0.509 | | 50.965233 | 0.455 |
| VSR-LKH | 12884 | 23.131658 | 0.105 | 25613 | 32.447953 | 0.200 | 103885 | 50.965300 | 0.468 |
| NeuroLKH | | **23.131414** | **0.000** | | **32.447304** | **0.000** | | **50.962916** | **0.000** |

In Figure 2, we plot the performance of the LKH, VSR-LKH and NeuroLKH algorithms for solving the testing datasets with different numbers of nodes against different running time to visualize the improvement process (the resulting objective values after each trial). The time limits are set to the longest ones used in Table 1 and Table 2, which are the running time of LKH with 1000 trials. Clearly, NeuroLKH outperforms both LKH and VSR-LKH significantly and consistently across different problem sizes and with different time limits. In particular, NeuroLKH is superior as it not only reaches good solutions fast but also converges to better solutions eventually. With the same performance (i.e. objective value), NeuroLKH considerably reduces the computational time. We can also conclude that when the time limit is short, the improvement of NeuroLKH over baselines is particularly substantial. In addition, we show that the subgradient optimization is necessary for LKH and VSR-LKH. As exhibited in Figure 2, the performances of both LKH and VSR-LKH are much worse without subgradient optimization (w/o SO). More impressively, even ignoring the preprocessing time (IPT) used for subgradient optimization (pertaining to LKH and VSR-LKH) and Sparse Graph Network inferring (pertaining to NeuroLKH), NeuroLKH still outstrips both LKH and VSR-LKH. Note that this comparison is unfair for NeuroLKH as LKH and VSR-LKH consume much longer preprocessing time which is unavoidable.

For the results reported in Table 1 and Table 2, almost all the improvements of NeuroLKH over LKH and VSR-LKH on different sizes and with different time limits are statistically significant with confidence levels larger than 99%. The only one exception is the performance on TSP with 100 nodes and the running time of LKH with 1000 trials where the confidence levels are 90.5% and 97.6% for the improvements, respectively.

In the Appendix Section A, we also show that NeuroLKH substantially outperforms other deep learning based methods [21, 20, 33, 18, 15, 8, 22, 5].

**Generalization to TSPLIB benchmark**. Besides generalization to larger sizes, generalization to different distributions remains a crucial challenge for deep learning based methods in existing works. The TSPLIB benchmark contains instances with various node distributions, making it extremely hard for such methods. We test on all the 72 TSPLIB instances with Euclidean distances and less than 10000 nodes. The number of trials is set to be the number of nodes and the algorithms are run 10 times for each instance following the convention for TSPLIB in [12, 36]. With the various unknown node distributions, we do not fine-tune the model for the node penalties and only use the edge scores in NeuroLKH. For the 24 instances labeled as hard in [36], which the original LKH fails to solve optimally during at least one of the 10 runs, NeuroLKH trained with uniformly distributed data is able to find optimal solutions 6.13 times on average, which is much better than LKH (3.75 times). As an active learning method, VSR-LKH finds optimal solutions 6.42 times on average, slightly better than NeuroLKH. While NeuroLKH improves the results on most hard instances, it could generalize poorly on instances with certain special patterns such as where most nodes are located along several horizontal lines, making it fail to solve 11 of the 48 easy instances optimally for some runs.

With the same training dataset size, we trained another model NeuroLKH_M using a mixture of instances with uniformly distributed nodes, clustered nodes with 3-8 clusters, half uniform and half clustered nodes following [30]. NeuroLKH_M finds optimal solutions 6.79 times on average for the hard instances and fails to solve only 5 easy instances optimally for some runs, better than the

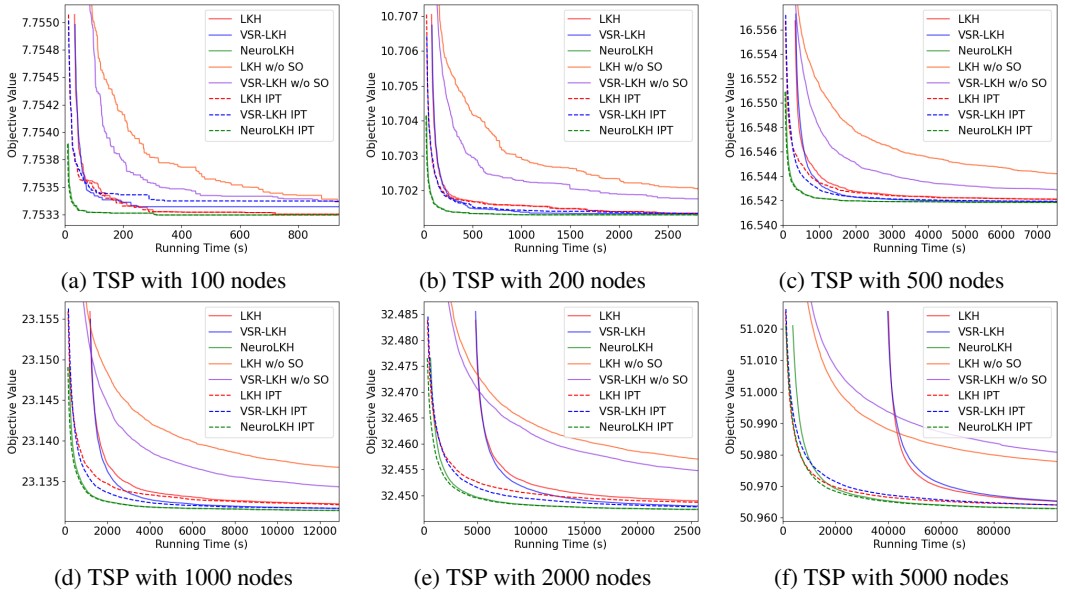

Figure 2: Performances of LKH, VSR-LKH and NeuroLKH for solving TSP with different sizes against different running time

NeuroLKH trained with only uniformly distributed instances. For all the 72 instances, NeuroLKH_M finds optimal solutions 8.74 times on average, which is much better than LKH (7.92 times) but slightly worse than VSR-LKH (8.78 times). Detailed results of each instance are listed in the Appendix Section B.

## 5.2 Experiments on other routing problems

Finally, we show that NeuroLKH can be easily extended to solve much more complicated routing problems such as the Capacitated Vehicle Routing Problem (CVRP), the Pickup and Delivery Problem (PDP) and CVRP with Time Windows (CVRPTW). We briefly introduce the problems in the Appendix Section C. Different from TSP, the node penalties do not apply to these problems. Therefore, NeuroLKH only learns the edge candidate set. As these three problems are very hard to solve and the optimal solutions are not available in a reasonable amount of time, we use LKH with 10000 trials to get solutions as training labels. The demands, capacities, starts and ends in the time windows are taken as node inputs along with the coordinates. For PDP, we add connections between each pair of pickup and delivery nodes and assign weight matrices for these connections in Eq. (2). For PDP and CVRPTW, the edge directions affect the tour feasibility therefore the model learns the in-direction edge scores and out-direction edge scores for each node with Eq. (5).

The node coordinates are also generated uniformly from the unit square for all three problems, following [21, 23]. For CVRP, the demands of customers are generated uniformly from integers $\{1..9\}$ with the capacity fixed as $40 + 0.1 \times |V|$, compatible with the largest CVRP (100 nodes) studied in [21]. For CVRPTW, we use the same way to generate demands, capacity, serving time and time windows as [7]. A training dataset for CVRP with 101-500 nodes and $120000/|V|$ instances for each size (about 180000 in total) is used to train the SGN for 10 epochs. PDP and CVRPTW are harder to solve therefore we use a training dataset with 41-200 nodes and $240000/|V|$ instances for each size. The other hyperparameters in SGN are the same as TSP and those for LKH searching process are consistent with example scripts given by LKH for CVRP, PDP and CVRPTW (with the SPECIAL hyperparameter).

In Table 3, we show the performance of NeuroLKH and the original LKH on testing datasets with 1000 instances for the smallest and largest graph sizes (number of customers) used in training as well as a much larger generalization size. We use the solving time of LKH with 100, 1000, 10000 trials as the time limits. For 100 trials, both methods fail to find feasible solutions for less than 1% of the PDP and CVRPTW test instances with 300 nodes. Whenever this happens, we push the infeasible visits to

Table 3: Comparative results for other routing problems

| | Method | Time(s) | Obj | Gap(%) | Time(s) | Obj | Gap(%) | Time(s) | Obj | Gap(%) |
|---|---|---|---|---|---|---|---|---|---|---|
| | | | $|V| = 100$ | | | $|V| = 500$ | | Generalization $|V| = 1000$ | | |
| CVRP | LKH (100 trials)
NeuroLKH | 485 | 15.8363
**15.7770** | 1.675
**1.295** | 2043 | 42.1621
**41.7311** | 5.394
**4.316** | 4607 | 58.1372
**56.6469** | 9.750
**6.937** |
| | LKH (1000 trials)
NeuroLKH | 4520 | 15.6483
**15.6295** | 0.468
**0.348** | 15812 | 40.6103
**40.4974** | 1.515
**1.233** | 30133 | 54.3412
**54.0499** | 2.584
**2.034** |
| | LKH (10000 trials)
NeuroLKH | 45435 | 15.5823
**15.5754** | 0.044
**0.000** | 166875 | 40.0670
**40.0043** | 0.157
**0.000** | 319368 | 53.1093
**52.9723** | 0.259
**0.000** |
| | | | $|V| = 40$ | | | $|V| = 200$ | | Generalization $|V| = 300$ | | |
| PDP | LKH (100 trials)
NeuroLKH | 115 | 6.2495
**6.2241** | 0.819
**0.409** | 2832 | 13.8390
**13.6246** | 5.535
**3.899** | 7939 | 17.0913
**16.7867** | 6.916
**5.011** |
| | LKH (1000 trials)
NeuroLKH | 845 | 6.2088
**6.2041** | 0.163
**0.087** | 21216 | 13.2850
**13.2443** | 1.310
**0.999** | 55643 | 16.2447
**16.1857** | 1.620
**1.251** |
| | LKH (10000 trials)
NeuroLKH | 7989 | 6.1998
**6.1988** | 0.018
**0.000** | 195220 | 13.1387
**13.1132** | 0.194
**0.000** | 515377 | 16.0119
**15.9857** | 0.163
**0.000** |
| | | | $|V| = 40$ | | | $|V| = 200$ | | Generalization $|V| = 300$ | | |
| CVRPTW | LKH (100 trials)
NeuroLKH | 147 | 9.3051
**9.2606** | 1.081
**0.597** | 813 | 26.1757
**25.4000** | 7.124
**3.949** | 1746 | 34.2301
**32.9676** | 8.798
**4.786** |
| | LKH (1000 trials)
NeuroLKH | 1017 | 9.2276
**9.2207** | 0.239
**0.164** | 4525 | 24.9770
**24.7857** | 2.218
**1.435** | 7820 | 32.2671
**32.0224** | 2.559
**1.781** |
| | LKH (10000 trials)
NeuroLKH | 9624 | 9.2073
**9.2056** | 0.018
**0.000** | 45509 | 24.5338
**24.4350** | 0.405
**0.000** | 75481 | 31.5719
**31.4620** | 0.350
**0.000** |

the end to get feasible solutions. The inferring time of SGN is 1s, 3s, 7s, 10s, 19s and 40s in total for the 1000 instances in the testing datasets with 40, 100, 200, 300, 500 and 1000 nodes, which is a tiny fraction compared to the LKH searching process. As shown in Table 3, NeuroLKH significantly improves the solution quality compared with the original LKH which is a very strong heuristic solver for all three problems, showing its potential in handling various types of routing problems.

**Performance on traditional benchmarks.** To show the effectiveness of NeuroLKH on complicated routing problems with various distributions, we perform experiments on CVRPLIB [30] and Solomon [28] benchmark datasets. CVRPLIB [30] contains various sized CVRP instances with a combination of 3 depot positioning, 3 customer positioning and 7 demand distributions. Solomon benchmark [28] contains CVRPTW instances with 100 customers and various distributions of time windows. We detail the benchmarks, the training datasets and the results for each instance in the Appendix Section D. In summary, tested on the 43 instances with 100-300 nodes in CVRPLIB [30], NeuroLKH improves the average performances on 38, 38 and 31 instances when the time limits are set to the time of LKH with 100, 1000 and 10000 trials, respectively. On the 11 Solomon R2-type instances, NeuroLKH outperforms LKH almost consistently with all the settings (32 out of the 33).

## 6    Conclusion

In this paper, we propose an algorithm utilizing the great power of deep learning models to combine with a strong heuristic for TSP. Specifically, one Sparse Graph Network is trained to predict the edge scores and the node penalties for generating the edge candidate set and transforming the edge distances, respectively. As shown in the extensive experiments, the improvement of NeuroLKH over baseline algorithms within different time limits is consistent and significant. And NeuroLKH generalizes well to instances with much larger graph sizes than training sizes and traditional benchmarks with various node distributions. Also, we use CVRP, PDP and CVRPTW to demonstrate that NeuroLKH effectively applies to other routing problems. NeuroLKH can effectively learn the routing patterns for TSP which generalize well to much larger sizes and different distributions of nodes. However, for other complicated routing problems such as CVRP and CVRPTW, although NeuroLKH generalizes well to larger sizes, it is hard to directly generalize to other distributions of demands and time windows without training, which is a limitation of NeuroLKH and is left for future research. In addition, NeuroLKH can be further combined with other learning based techniques such as sparsifying the TSP graph [29] and other strong traditional algorithms such as the Hybrid Genetic Search [31].

## Acknowledgments and Disclosure of Funding

This work was supported by the A*STAR Cyber-Physical Production System (CPPS) – Towards Contextual and Intelligent Response Research Program, under the RIE2020 IAF-PP Grant A19C1a0018, and Model Factory@SIMTech, in part by the National Natural Science Foundation of China under Grant 61803104 and Grant 62102228, and in part by the Young Scholar Future Plan of Shandong University under Grant 62420089964188.

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
