# NeuroLKH: Combining Deep Learning Model with Lin-Kernighan-Helsgaun Heuristic for Solving the Traveling Salesman Problem (Appendix)

## A    Experiments for TSP

To verify the quality of the edge candidate set learned by NeuroLKH, we report two metrics for the edge candidate set attained by different methods, i.e., the average ranking of the optimal edges and the percentage of optimal edges missed in the set, respectively. Regarding the sensitivity analysis of the Minimum Spanning Tree with the subgradient optimization in LKH algorithm, 0.68% and 0.67% of the optimal edges are missed in the candidate set for TSP100 and TSP500, respectively, where the average rankings of optimal edges are 1.670 and 1.681. The ideal average ranking would be 1.5 since the two optimal edges for each node would be the first and the second in the ranks. NeuroLKH reduces the average ranking to 1.557 and 1.597 where only 0.05% and 0.09% of the optimal edges are missed in the set, which justifies the effectiveness of NeuroLKH in learning desirable edge candidates.

For TSP, we choose the number of directed edges pointed from one node in the sparse edge set $E^*$ as $\gamma = 20$ to include most of the edges in the optimal tours into the sparse graph, which results in only 0.01% of the optimal edges missed in the sparse graph for the training dataset. In our experiments with $\gamma = 10, 20, 30$ (trained with 20% of the training samples to save time), 0.643%, 0.209% and 0.208% of the optimal edges are missed in the candidate set with the average ranking of the optimal edges 1.653, 1.646 and 1.640 for TSP500, respectively. With $\gamma > 20$ (i.e. numbers of edges), it only improves the average ranking marginally with similar percentages of optimal edges but obviously increases the computational time. Pertaining to other routing problems, we find similar results therefore we use $\gamma = 20$ for consistency. We find that the network can hardly give a high edge score to an edge with considerably large Euclidean distance and include it into the candidate set. Therefore larger $\gamma$ is not needed which does not impact the performance much as long as it is not too small (e.g. less than 20).

The model outputs the node penalties within the range of $[-C, C]$ with $C = 10$. In the original LKH algorithm, a subgradient optimization process is used to optimize the node penalties iteratively until convergence for each instance. In this process for the training instances where the coordinates are always between 0 and 1, we find that the penalties are usually between -10 and 10 (for different sizes). While testing for instances with different coordinate ranges, we scale the instances to make the coordinates between 0 and 1. The aspect ratio is fixed so that the objective value is just scaled by a constant. Therefore, we use $C = 10$ in our experiments.

In Table S.1, we compare NeuroLKH with other recently proposed Deep Learning based methods on TSP100. Notably, most of them can hardly handle problems with more than 100 nodes. One exception is the method in [8], which is tested on large problems but the performance deteriorates rapidly with the increase of problem size and is still inferior to LKH. We adopt the results from their original works where the datasets tested on might be different but are sampled from the same distribution. Therefore the optimality gap is a more important measure than the objective value. The running time is reported for solving 1000 instances in total with the assumption that it is linearly related to the number of instances. Apparently, NeuroLKH significantly outperforms other methods with a short running time. And more importantly, as shown in Table 1 and Table 2, NeuroLKH generalizes well to large TSP with up to 5000 nodes.

35th Conference on Neural Information Processing Systems (NeurIPS 2021).

Table S.1: Comparative results on TSP100. Here we report three results of NeuroLKH with different time limits from Table 1.

| Method | Time(s) | Gap(‰) | Method | Time(s) | Gap(‰) | Method | Time(s) | Gap(‰) |
|---|---|---|---|---|---|---|---|---|
| GCN greedy [18] | 36 | 838.000 | AM Greedy [21] | 0.6 | 453.000 | AM sampling [21] | 360 | 226.000 |
| Wu [33] | 720 | 142.000 | GCN bs [18] | 240 | 139.000 | CVAE-Opt-RS [15] | 50500 | 135.000 |
| da Costa [5] | 246 | 87.000 | CVAE-Opt-DE [15] | 55100 | 34.000 | POMO [22] | 6 | 14.000 |
| Fu [8] | 90 | 4.000 | DPDP 10k [20] | 456 | 0.900 | DPDP 100k [20] | 990 | 0.400 |
| **NeuroLKH** | 33 | 0.111 | **NeuroLKH** | 127 | 0.030 | **NeuroLKH** | 938 | 0.000 |

# B  Experiments for TSPLIB

NeuroLKH is trained using only the instances with nodes generated from the uniform distribution. With the same training dataset size, we trained another model NeuroLKH_M using a mixture of instances with uniformly distributed nodes, clustered nodes with 3-8 clusters, half uniform and half clustered nodes following [30]. Following the convention for TSPLIB in [12, 36], the number of trials is set to be the number of nodes and the algorithms are run 10 times for each instance. During each run, the algorithm will stop when the optimal solution is found and the number of trials actually conducted is reported. Here we show the results of LKH, VSR-LKH, NeuroLKH and NeuroLKH_M for each instance in Table S.2, Table S.3 and Table S.4. The optimal tour distance is shown under the instance name. We report the success times where the optimal solution is found, the best performance (tour distance) during the runs, the average performance, the average running time (seconds) and the average number of trials actually conducted. The results of LKH are the same as reported in [36] (except the running time where we run all the algorithms on our machine for a fair comparison) while the results of VSR-LKH are slightly different due to behaviour uncontrolled by the random seed in the code.

# C  Experiments for Other Routing Problems

Here we briefly introduce the Capacitated Vehicle Routing Problem (CVRP), the Pickup and Delivery Problem (PDP) and CVRP with Time Windows (CVRPTW). For PDP, the customers contain pairs of pickup and delivery nodes. The vehicle starts from the depot, visits each customer node once and returns to the depot with the constraint that the pickup node must be visited before the corresponding delivery node. For CVRP, multiple routes can be planned. In each route, the vehicle starts from the depot, visits some customers and returns to the depot. The total demand of the customers in each route cannot exceed the vehicle capacity and each customer must be visited once. CVRPTW generalizes CVRP with an additional constraint that each customer must be visited within the corresponding time window. The time will be spent on traveling between the nodes and serving the customers. The goal of all three problems is to minimize the tour distance.

Similarly, we plot the performance of the LKH and NeuroLKH algorithms for solving CVRP, PDP and CVRPTW in Figure S.1, which shows similar trends as those in Figure 2. The time limits are set to the longest ones used in Table 3, i.e., the running time of LKH algorithm with 10000 trials.

For the results reported in Table 3, almost all the improvements of NeuroLKH over LKH on different sizes and with different time limits are statistically significant with confidence levels larger than 99%. The only exceptions are the performance for the smallest size of each problem and the longest time limits (the running time of LKH with 10000 trials), where the confidence levels are 98.7%, 98.9% and 77.9% for CVRP100, PDP40 and CVRPTW40, respectively. The confidence level for CVRPTW40 with the time limit of LKH with 10000 trials is relatively low because CVRPTW with 40 nodes solved by LKH is already fairly close to the optimality with such a long time limit. Therefore the improvement room left for NeuroLKH is small.

# D  Experiments on CVRPLIB and Solomon Benchmark

CVRPLIB [30] contains various sized CVRP instances with a combination of 3 depot positioning, 3 customer positioning and 7 demand distributions. We train one network using CVRP instances ranging from 101 to 300 nodes. The instances are generated from this mixture of distributions

proposed in [30] and we generate $120000/|V|$ instances for each size $|V|$ in the training dataset, resulting in approximately 120000 instances in total.

Solomon benchmark [28] contains CVRPTW instances with 100 customers and various distributions of time windows. An additional constraint for this benchmark is to minimize the number of routes. Therefore the goal is to minimize the tour distance using the minimum number of routes. We choose R2-type as the testbed in our experiment. We generate a training dataset of instances with 100 customers. The node coordinates are generated independently from the uniform distribution ranging from 0 to 80. The demands are generated from a Gaussian distribution with mean 15 and standard deviation 10 and the capacity is fixed as 1000. The serving time $s$ for each customer is fixed as 10. The center of time window for node $i$ is generated from the uniform distribution with the interval $[dist, 1000 - s - dist]$, where $dist$ is the distance between node $i$ and the depot. And the width of time window is generated from a Gaussian distribution with the mean and standard deviation set to 115 and 35, 240 and 0, 350 and 160, 150 and 380, 470 and 70, respectively. For each of the first two sets of parameters, four different types are generated with 0%, 25%, 50% and 100% of the customers receiving the time windows. And for the last three sets of parameters, all customers are receiving the time windows, resulting in 11 types of instances in total. We generate 5000 instances for each type in the training dataset. Please refer to the code for more details.

As the running time is all relatively short, we run both LKH and NeuroLKH for 100 times on each instance. The results of LKH and NeuroLKH are shown in Table S.5, Table S.6 and Table S.7, while the time limits are set to the running time of LKH with 100, 1000 and 10000 trials. The optimal tour distance is shown under the instance name. We report the average running time (seconds), the best performance (tour distance) during the runs, the average performance, the success times when the optimal solution is found.

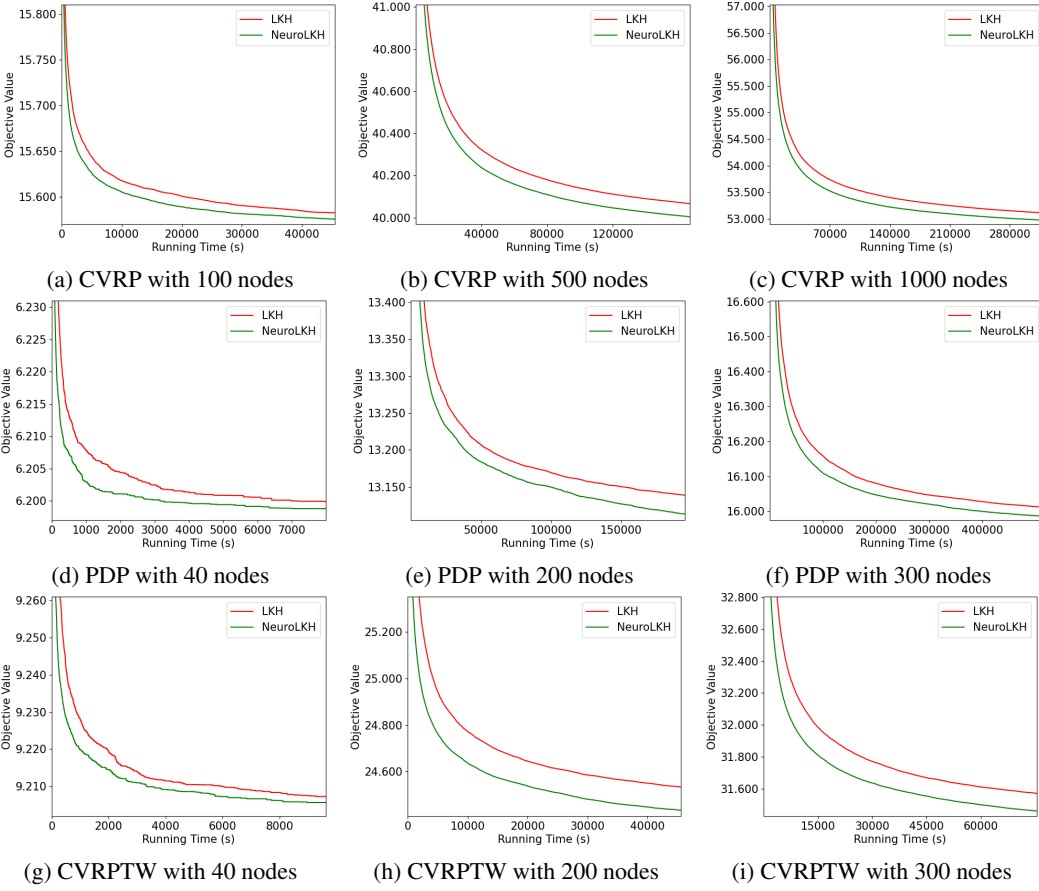

Figure S.1: Performances of LKH and NeuroLKH for solving CVRP, PDP and CVRPTW with different sizes against different running time

Table S.2: TSPLIB results for each hard instance

| Method | Name | Success | Best | Average | Time | Trials | Name | Success | Best | Average | Time | Trials |
|---|---|---|---|---|---|---|---|---|---|---|---|---|
| LKH | kroB150 | 2/10 | 26130 | 26131.6 | 0.32 | 128.4 | rat195 | 9/10 | 2323 | 2323.5 | 0.22 | 55 |
| VSR-LKH | | 4/10 | 26130 | 26131.2 | 0.21 | 106.3 | | 9/10 | 2323 | 2323.5 | 0.36 | 69.5 |
| NeuroLKH_R | 26130 | 10/10 | 26130 | 26130 | 0.07 | 9.8 | 2323 | 10/10 | 2323 | 2323 | 0.11 | 8.4 |
| NeuroLKH_M | | 10/10 | 26130 | 26130 | 0.12 | 22.1 | | 10/10 | 2323 | 2323 | 0.06 | 3.9 |
| LKH | pr299 | 9/10 | 48191 | 48194.3 | 0.4 | 51.7 | d493 | 6/10 | 35002 | 35002.8 | 4.71 | 219.6 |
| VSR-LKH | | 10/10 | 48191 | 48191 | 0.43 | 13.6 | | 10/10 | 35002 | 35002 | 0.5 | 8.8 |
| NeuroLKH_R | 48191 | 10/10 | 48191 | 48191 | 0.25 | 10.1 | 35002 | 6/10 | 35002 | 35032.2 | 6.73 | 320.5 |
| NeuroLKH_M | | 10/10 | 48191 | 48191 | 0.22 | 13.2 | | 10/10 | 35002 | 35002 | 0.67 | 27.5 |
| LKH | rat575 | 2/10 | 6773 | 6773.8 | 3.23 | 526.9 | pr1002 | 8/10 | 259045 | 259045.6 | 4.53 | 549 |
| VSR-LKH | | 6/10 | 6773 | 6773.4 | 3.2 | 310.6 | | 10/10 | 259045 | 259045 | 0.72 | 16 |
| NeuroLKH_R | 6773 | 9/10 | 6773 | 6773.1 | 1.91 | 179 | 259045 | 10/10 | 259045 | 259045 | 8.46 | 330.6 |
| NeuroLKH_M | | 7/10 | 6773 | 6773.3 | 3.87 | 345.3 | | 10/10 | 259045 | 259045 | 1.05 | 34 |
| LKH | u1060 | 5/10 | 224094 | 224107.5 | 101.76 | 663.3 | vm1084 | 3/10 | 239297 | 239372.6 | 46.16 | 824.1 |
| VSR-LKH | | 10/10 | 224094 | 224094 | 3.52 | 19.1 | | 7/10 | 239297 | 239312.6 | 49.41 | 474.8 |
| NeuroLKH_R | 224094 | 10/10 | 224094 | 224094 | 35.07 | 206.9 | 239297 | 1/10 | 239297 | 239379.5 | 23.4 | 1028.9 |
| NeuroLKH_M | | 10/10 | 224094 | 224094 | 10.05 | 75.4 | | 7/10 | 239297 | 239315.1 | 21.29 | 439.7 |
| LKH | pcb1173 | 4/10 | 56892 | 56895 | 5.37 | 844 | rl1304 | 3/10 | 252948 | 253156.4 | 18.28 | 1170 |
| VSR-LKH | | 8/10 | 56892 | 56893 | 7.07 | 436.9 | | 10/10 | 252948 | 252948 | 1.44 | 17.9 |
| NeuroLKH_R | 56892 | 9/10 | 56892 | 56892.5 | 5.32 | 410.4 | 252948 | 9/10 | 252948 | 252953.1 | 9.26 | 370.8 |
| NeuroLKH_M | | 8/10 | 56892 | 56893 | 6.48 | 378.2 | | 8/10 | 252948 | 252958.2 | 11.36 | 600.6 |
| LKH | rl1323 | 6/10 | 270199 | 270219.6 | 12.57 | 718.8 | nrw1379 | 6/10 | 56638 | 56640 | 9.84 | 759.3 |
| VSR-LKH | | 10/10 | 270199 | 270199 | 9.08 | 189.7 | | 9/10 | 56638 | 56638.5 | 12.84 | 253.7 |
| NeuroLKH_R | 270199 | 7/10 | 270199 | 270247.9 | 16.59 | 742.2 | 56638 | 9/10 | 56638 | 56638.5 | 15.28 | 372.4 |
| NeuroLKH_M | | 8/10 | 270199 | 270204.4 | 11.13 | 538.5 | | 10/10 | 56638 | 56638 | 7.85 | 260.8 |
| LKH | fl1400 | 1/10 | 20127 | 20160.3 | 2703.75 | 1372.9 | fl1577 | 0/10 | 22254 | 22260.6 | 965.98 | 1577 |
| VSR-LKH | | 1/10 | 20127 | 20160.3 | 3323.31 | 1380.6 | | 0/10 | 22254 | 22255.8 | 3095.13 | 1577 |
| NeuroLKH_R | 20127 | 0/10 | 20165 | 20235.5 | 356.77 | 1400 | 22249 | 1/10 | 22249 | 22256.6 | 652.75 | 1445.8 |
| NeuroLKH_M | | 0/10 | 20164 | 20169.4 | 754.03 | 1400 | | 0/10 | 22254 | 22302.8 | 522.49 | 1577 |
| LKH | vm1748 | 9/10 | 336556 | 336557.3 | 17.62 | 1007.9 | u1817 | 1/10 | 57201 | 57251.1 | 63.28 | 1817 |
| VSR-LKH | | 10/10 | 336556 | 336556 | 5.42 | 37.8 | | 7/10 | 57201 | 57212 | 159.43 | 967 |
| NeuroLKH_R | 336556 | 5/10 | 336556 | 336628 | 38.16 | 1282.9 | 57201 | 2/10 | 57201 | 57227.3 | 238.86 | 1803.4 |
| NeuroLKH_M | | 10/10 | 336556 | 336556 | 13.65 | 460.2 | | 2/10 | 57201 | 57225.2 | 126.01 | 1691.5 |
| LKH | rl1889 | 0/10 | 316549 | 316549.8 | 59.31 | 1889 | d2103 | 0/10 | 80454 | 80462 | 111.69 | 2103 |
| VSR-LKH | | 4/10 | 316536 | 316569 | 143.58 | 1393.9 | | 0/10 | 80454 | 80454.2 | 619.38 | 2103 |
| NeuroLKH_R | 316536 | 0/10 | 316638 | 316648.7 | 141.23 | 1889 | 80450 | 4/10 | 80450 | 80452.4 | 339.12 | 1560.3 |
| NeuroLKH_M | | 3/10 | 316536 | 316619.4 | 81.93 | 1485.6 | | 3/10 | 80450 | 80454.6 | 213 | 1614.7 |
| LKH | u2152 | 3/10 | 64253 | 64287.7 | 88.79 | 1614 | pcb3038 | 4/10 | 137694 | 137701.2 | 79.22 | 2078.6 |
| VSR-LKH | | 7/10 | 64253 | 64270.1 | 178.54 | 1334.7 | | 7/10 | 137694 | 137695.5 | 214.24 | 1422.2 |
| NeuroLKH_R | 64253 | 9/10 | 64253 | 64258.7 | 56.63 | 520.9 | 137694 | 8/10 | 137694 | 137695 | 151.91 | 1104 |
| NeuroLKH_M | | 8/10 | 64253 | 64255.2 | 66.85 | 878.1 | | 8/10 | 137694 | 137695 | 99.23 | 1084.6 |
| LKH | fl3795 | 0/10 | 28813 | 28813.7 | 34045.95 | 3795 | fnl4461 | 9/10 | 182566 | 182566.5 | 31.89 | 923.1 |
| VSR-LKH | | 0/10 | 28831 | 28831 | 75405 | 3795 | | 10/10 | 182566 | 182566 | 19.94 | 89.1 |
| NeuroLKH_R | 28772 | 0/10 | 28999 | 29010.6 | 80797.24 | 3795 | 182566 | 10/10 | 182566 | 182566 | 27.91 | 171.5 |
| NeuroLKH_M | | 0/10 | 29488 | 29495.3 | 1329.72 | 3795 | | 10/10 | 182566 | 182566 | 19.26 | 151.5 |
| LKH | rl5915 | 0/10 | 565544 | 565581.2 | 221.29 | 5915 | rl5934 | 0/10 | 556136 | 556309.8 | 371.79 | 5934 |
| VSR-LKH | | 1/10 | 565530 | 565580.8 | 896.59 | 5354.9 | | 4/10 | 556045 | 556099.6 | 923.66 | 4804.7 |
| NeuroLKH_R | 565530 | 0/10 | 565585 | 565969.9 | 658.32 | 5915 | 556045 | 8/10 | 556045 | 556059.5 | 376.57 | 3470.2 |
| NeuroLKH_M | | 1/10 | 565530 | 565579.5 | 365.82 | 5352.9 | | 10/10 | 556045 | 556045 | 143.34 | 1529.8 |

Table S.3: TSPLIB results for each easy instance

| Method | Name | Success | Best | Average | Time | Trials | Name | Success | Best | Average | Time | Trials |
|---|---|---|---|---|---|---|---|---|---|---|---|---|
| LKH | eil51 | 10/10 | 426 | 426 | 0 | 1 | berlin52 | 10/10 | 7542 | 7542 | 0.01 | 0 |
| VSR-LKH | | 10/10 | 426 | 426 | 0 | 1 | | 10/10 | 7542 | 7542 | 0.02 | 0 |
| NeuroLKH_R | 426 | 10/10 | 426 | 426 | 0 | 1 | 7542 | 10/10 | 7542 | 7542 | 0.02 | 0 |
| NeuroLKH_M | | 10/10 | 426 | 426 | 0 | 1 | | 10/10 | 7542 | 7542 | 0.02 | 0 |
| LKH | st70 | 10/10 | 675 | 675 | 0.01 | 1 | eil76 | 10/10 | 538 | 538 | 0 | 1 |
| VSR-LKH | | 10/10 | 675 | 675 | 0.01 | 1 | | 10/10 | 538 | 538 | 0 | 1 |
| NeuroLKH_R | 675 | 10/10 | 675 | 675 | 0.01 | 1 | 538 | 10/10 | 538 | 538 | 0 | 1 |
| NeuroLKH_M | | 10/10 | 675 | 675 | 0.01 | 1 | | 10/10 | 538 | 538 | 0 | 1 |
| LKH | pr76 | 10/10 | 108159 | 108159 | 0.02 | 1 | rat99 | 10/10 | 1211 | 1211 | 0 | 1 |
| VSR-LKH | | 10/10 | 108159 | 108159 | 0.02 | 1 | | 10/10 | 1211 | 1211 | 0 | 1 |
| NeuroLKH_R | 108159 | 10/10 | 108159 | 108159 | 0.02 | 1 | 1211 | 10/10 | 1211 | 1211 | 0.01 | 1 |
| NeuroLKH_M | | 10/10 | 108159 | 108159 | 0.02 | 1 | | 10/10 | 1211 | 1211 | 0 | 1 |
| LKH | kroA100 | 10/10 | 21282 | 21282 | 0.02 | 1 | kroB100 | 10/10 | 22141 | 22141 | 0.03 | 1.2 |
| VSR-LKH | | 10/10 | 21282 | 21282 | 0.01 | 1 | | 10/10 | 22141 | 22141 | 0.04 | 2.5 |
| NeuroLKH_R | 21282 | 10/10 | 21282 | 21282 | 0.01 | 1 | 22141 | 10/10 | 22141 | 22141 | 0.03 | 1 |
| NeuroLKH_M | | 10/10 | 21282 | 21282 | 0.01 | 1 | | 10/10 | 22141 | 22141 | 0.03 | 1 |
| LKH | kroC100 | 10/10 | 20749 | 20749 | 0.01 | 1 | kroD100 | 10/10 | 21294 | 21294 | 0.02 | 1.8 |
| VSR-LKH | | 10/10 | 20749 | 20749 | 0.02 | 1 | | 10/10 | 21294 | 21294 | 0.02 | 1 |
| NeuroLKH_R | 20749 | 10/10 | 20749 | 20749 | 0.02 | 1 | 21294 | 10/10 | 21294 | 21294 | 0.02 | 1 |
| NeuroLKH_M | | 10/10 | 20749 | 20749 | 0.02 | 1 | | 10/10 | 21294 | 21294 | 0.02 | 1 |
| LKH | kroE100 | 10/10 | 22068 | 22068 | 0.03 | 3.2 | rd100 | 10/10 | 7910 | 7910 | 0 | 1 |
| VSR-LKH | | 10/10 | 22068 | 22068 | 0.06 | 8.5 | | 10/10 | 7910 | 7910 | 0 | 1 |
| NeuroLKH_R | 22068 | 10/10 | 22068 | 22068 | 0.03 | 1 | 7910 | 10/10 | 7910 | 7910 | 0.01 | 1 |
| NeuroLKH_M | | 10/10 | 22068 | 22068 | 0.04 | 4.8 | | 10/10 | 7910 | 7910 | 0.01 | 1 |
| LKH | eil101 | 10/10 | 629 | 629 | 0 | 1 | lin105 | 10/10 | 14379 | 14379 | 0 | 1 |
| VSR-LKH | | 10/10 | 629 | 629 | 0 | 1 | | 10/10 | 14379 | 14379 | 0 | 1 |
| NeuroLKH_R | 629 | 10/10 | 629 | 629 | 0 | 1 | 14379 | 10/10 | 14379 | 14379 | 0 | 1 |
| NeuroLKH_M | | 10/10 | 629 | 629 | 0 | 1 | | 10/10 | 14379 | 14379 | 0 | 1 |
| LKH | pr107 | 10/10 | 44303 | 44303 | 0.13 | 1 | pr124 | 10/10 | 59030 | 59030 | 0.04 | 1 |
| VSR-LKH | | 10/10 | 44303 | 44303 | 0.13 | 1 | | 10/10 | 59030 | 59030 | 0.04 | 1 |
| NeuroLKH_R | 44303 | 10/10 | 44303 | 44303 | 0.14 | 1.1 | 59030 | 10/10 | 59030 | 59030 | 0.07 | 1 |
| NeuroLKH_M | | 10/10 | 44303 | 44303 | 0.13 | 1 | | 10/10 | 59030 | 59030 | 0.06 | 1 |
| LKH | bier127 | 10/10 | 118282 | 118282 | 0.01 | 1 | ch130 | 10/10 | 6110 | 6110 | 0.03 | 1 |
| VSR-LKH | | 10/10 | 118282 | 118282 | 0.02 | 1 | | 10/10 | 6110 | 6110 | 0.07 | 7.3 |
| NeuroLKH_R | 118282 | 4/10 | 118282 | 118300.6 | 0.13 | 102.5 | 6110 | 10/10 | 6110 | 6110 | 0.02 | 1.1 |
| NeuroLKH_M | | 10/10 | 118282 | 118282 | 0.01 | 1 | | 10/10 | 6110 | 6110 | 0.03 | 2.1 |
| LKH | pr136 | 10/10 | 96772 | 96772 | 0.08 | 1 | pr144 | 10/10 | 58537 | 58537 | 0.37 | 1 |
| VSR-LKH | | 10/10 | 96772 | 96772 | 0.08 | 1 | | 10/10 | 58537 | 58537 | 0.43 | 1 |
| NeuroLKH_R | 96772 | 10/10 | 96772 | 96772 | 0.15 | 4.5 | 58537 | 1/10 | 58537 | 58584.7 | 2.6 | 131.8 |
| NeuroLKH_M | | 10/10 | 96772 | 96772 | 0.11 | 1 | | 2/10 | 58537 | 58614 | 2.31 | 122.3 |
| LKH | ch150 | 10/10 | 6528 | 6528 | 0.04 | 1.7 | kroA150 | 10/10 | 26524 | 26524 | 0.05 | 3.8 |
| VSR-LKH | | 10/10 | 6528 | 6528 | 0.02 | 1 | | 10/10 | 26524 | 26524 | 0.04 | 1 |
| NeuroLKH_R | 6528 | 10/10 | 6528 | 6528 | 0.02 | 1.1 | 26524 | 10/10 | 26524 | 26524 | 0.04 | 2.6 |
| NeuroLKH_M | | 10/10 | 6528 | 6528 | 0.02 | 1.1 | | 10/10 | 26524 | 26524 | 0.02 | 1 |
| LKH | pr152 | 10/10 | 73682 | 73682 | 0.48 | 29.4 | u159 | 10/10 | 42080 | 42080 | 0.01 | 1 |
| VSR-LKH | | 8/10 | 73682 | 73709.2 | 0.69 | 47 | | 10/10 | 42080 | 42080 | 0.01 | 1 |
| NeuroLKH_R | 73682 | 8/10 | 73682 | 73709.2 | 1.44 | 59.6 | 42080 | 10/10 | 42080 | 42080 | 0.01 | 1 |
| NeuroLKH_M | | 9/10 | 73682 | 73695.6 | 0.87 | 38.7 | | 10/10 | 42080 | 42080 | 0.01 | 1 |

Table S.4: TSPLIB results for each easy instance (continued)

| Method | Name | Success | Best | Average | Time | Trials | Name | Success | Best | Average | Time | Trials |
|---|---|---|---|---|---|---|---|---|---|---|---|---|
| LKH | d198 | 10/10 | 15780 | 15780 | 0.57 | 1 | kroA200 | 10/10 | 29368 | 29368 | 0.06 | 1.7 |
| VSR-LKH | | 10/10 | 15780 | 15780 | 0.43 | 1 | | 10/10 | 29368 | 29368 | 0.06 | 1.5 |
| NeuroLKH_R | 15780 | 0/10 | 15789 | 15825 | 2.54 | 198 | 29368 | 10/10 | 29368 | 29368 | 0.05 | 1 |
| NeuroLKH_M | | 10/10 | 15780 | 15780 | 0.87 | 1 | | 10/10 | 29368 | 29368 | 0.04 | 1 |
| LKH | kroB200 | 10/10 | 29437 | 29437 | 0.02 | 1 | ts225 | 10/10 | 126643 | 126643 | 0.04 | 1 |
| VSR-LKH | | 10/10 | 29437 | 29437 | 0.03 | 1 | | 10/10 | 126643 | 126643 | 0.02 | 1 |
| NeuroLKH_R | 29437 | 10/10 | 29437 | 29437 | 0.02 | 1 | 126643 | 10/10 | 126643 | 126643 | 0.06 | 1 |
| NeuroLKH_M | | 10/10 | 29437 | 29437 | 0.02 | 1 | | 10/10 | 126643 | 126643 | 0.06 | 1 |
| LKH | tsp225 | 10/10 | 3916 | 3916 | 0.06 | 1 | pr226 | 10/10 | 80369 | 80369 | 0.08 | 1 |
| VSR-LKH | | 10/10 | 3916 | 3916 | 0.07 | 1 | | 10/10 | 80369 | 80369 | 0.1 | 13.3 |
| NeuroLKH_R | 3916 | 10/10 | 3916 | 3916 | 0.06 | 1 | 80369 | 6/10 | 80369 | 80381.7 | 1.34 | 146.2 |
| NeuroLKH_M | | 10/10 | 3916 | 3916 | 0.06 | 1 | | 10/10 | 80369 | 80369 | 0.22 | 5.9 |
| LKH | gil262 | 10/10 | 2378 | 2378 | 0.14 | 10.6 | pr264 | 10/10 | 49135 | 49135 | 0.24 | 14.4 |
| VSR-LKH | | 10/10 | 2378 | 2378 | 0.05 | 1.7 | | 10/10 | 49135 | 49135 | 0.19 | 1 |
| NeuroLKH_R | 2378 | 10/10 | 2378 | 2378 | 0.13 | 8 | 49135 | 10/10 | 49135 | 49135 | 0.13 | 6.2 |
| NeuroLKH_M | | 10/10 | 2378 | 2378 | 0.05 | 2.2 | | 10/10 | 49135 | 49135 | 0.09 | 2.4 |
| LKH | a280 | 10/10 | 2579 | 2579 | 0.03 | 1 | lin318 | 10/10 | 42029 | 42029 | 0.23 | 27.9 |
| VSR-LKH | | 10/10 | 2579 | 2579 | 0.02 | 1 | | 10/10 | 42029 | 42029 | 0.09 | 1.8 |
| NeuroLKH_R | 2579 | 10/10 | 2579 | 2579 | 0.02 | 1 | 42029 | 10/10 | 42029 | 42029 | 0.18 | 3.6 |
| NeuroLKH_M | | 10/10 | 2579 | 2579 | 0.03 | 1 | | 10/10 | 42029 | 42029 | 0.15 | 5.9 |
| LKH | rd400 | 10/10 | 15281 | 15281 | 0.23 | 33 | fl417 | 10/10 | 11861 | 11861 | 2.69 | 7.3 |
| VSR-LKH | | 10/10 | 15281 | 15281 | 0.23 | 11.6 | | 10/10 | 11861 | 11861 | 1.91 | 3.7 |
| NeuroLKH_R | 15281 | 10/10 | 15281 | 15281 | 0.11 | 3.9 | 11861 | 5/10 | 11861 | 11867.6 | 16.64 | 337.2 |
| NeuroLKH_M | | 10/10 | 15281 | 15281 | 0.12 | 4.7 | | 9/10 | 11861 | 11861.1 | 16.7 | 51.7 |
| LKH | pr439 | 10/10 | 107217 | 107217 | 0.59 | 39.5 | pcb442 | 10/10 | 50778 | 50778 | 0.16 | 8.2 |
| VSR-LKH | | 10/10 | 107217 | 107217 | 0.44 | 22.1 | | 10/10 | 50778 | 50778 | 0.07 | 3 |
| NeuroLKH_R | 107217 | 3/10 | 107217 | 107267.4 | 1.64 | 320.1 | 50778 | 10/10 | 50778 | 50778 | 0.11 | 3.8 |
| NeuroLKH_M | | 9/10 | 107217 | 107224.2 | 0.71 | 90.3 | | 10/10 | 50778 | 50778 | 0.18 | 6.9 |
| LKH | u574 | 10/10 | 36905 | 36905 | 0.8 | 149.9 | p654 | 10/10 | 34643 | 34643 | 7.04 | 22.9 |
| VSR-LKH | | 10/10 | 36905 | 36905 | 0.39 | 29.2 | | 10/10 | 34643 | 34643 | 4.28 | 9 |
| NeuroLKH_R | 36905 | 10/10 | 36905 | 36905 | 0.2 | 3.8 | 34643 | 1/10 | 34643 | 34765.8 | 40.27 | 619 |
| NeuroLKH_M | | 10/10 | 36905 | 36905 | 0.11 | 1.9 | | 10/10 | 34643 | 34643 | 2.63 | 7 |
| LKH | d657 | 10/10 | 48912 | 48912 | 0.48 | 33.5 | u724 | 10/10 | 41910 | 41910 | 1.53 | 125.4 |
| VSR-LKH | | 10/10 | 48912 | 48912 | 0.44 | 21 | | 10/10 | 41910 | 41910 | 0.85 | 23.3 |
| NeuroLKH_R | 48912 | 5/10 | 48912 | 48912.5 | 6.65 | 511.5 | 41910 | 10/10 | 41910 | 41910 | 0.94 | 46.6 |
| NeuroLKH_M | | 10/10 | 48912 | 48912 | 0.39 | 10 | | 10/10 | 41910 | 41910 | 0.64 | 16.8 |
| LKH | rat783 | 10/10 | 8806 | 8806 | 0.08 | 4.2 | d1291 | 10/10 | 50801 | 50801 | 6.27 | 192.1 |
| VSR-LKH | | 10/10 | 8806 | 8806 | 0.11 | 3.9 | | 10/10 | 50801 | 50801 | 2.51 | 39.5 |
| NeuroLKH_R | 8806 | 10/10 | 8806 | 8806 | 0.14 | 4.2 | 50801 | 9/10 | 50801 | 50803.4 | 5.64 | 274.4 |
| NeuroLKH_M | | 10/10 | 8806 | 8806 | 0.21 | 12.2 | | 7/10 | 50801 | 50808.2 | 9.46 | 437.4 |
| LKH | u1432 | 10/10 | 152970 | 152970 | 0.43 | 5.3 | d1655 | 10/10 | 62128 | 62128 | 5.44 | 176 |
| VSR-LKH | | 10/10 | 152970 | 152970 | 0.55 | 5 | | 10/10 | 62128 | 62128 | 0.94 | 9.8 |
| NeuroLKH_R | 152970 | 10/10 | 152970 | 152970 | 0.56 | 7.1 | 62128 | 8/10 | 62128 | 62128.2 | 22.22 | 870.4 |
| NeuroLKH_M | | 10/10 | 152970 | 152970 | 0.43 | 3.8 | | 10/10 | 62128 | 62128 | 7.86 | 214.1 |
| LKH | u2319 | 10/10 | 234256 | 234256 | 0.46 | 3.1 | pr2392 | 10/10 | 378032 | 378032 | 0.4 | 5.8 |
| VSR-LKH | | 10/10 | 234256 | 234256 | 0.89 | 3.9 | | 10/10 | 378032 | 378032 | 0.78 | 8.7 |
| NeuroLKH_R | 234256 | 10/10 | 234256 | 234256 | 0.67 | 3.5 | 378032 | 10/10 | 378032 | 378032 | 1.22 | 25.5 |
| NeuroLKH_M | | 10/10 | 234256 | 234256 | 0.37 | 2.6 | | 10/10 | 378032 | 378032 | 1.31 | 25.9 |

Table S.5: CVRPLIB results

| Name | Method | LKH with 100 trials as time limit | | | | LKH with 1000 trials as time limit | | | | LKH with 10000 trials as time limit | | | |
|---|---|---|---|---|---|---|---|---|---|---|---|---|---|
| | | Time | Best | Avg | Suc | Time | Best | Avg | Suc | Time | Best | Avg | Suc |
| X-n101-k25 | LKH | 1.2 | 27744 | 28214.5 | 0 | 13 | 27591 | 27794.3 | 6 | 131 | 27591 | **27667.0** | 33 |
| 27591 | NeuroLKH | | 27665 | **28146.6** | 0 | | 27591 | **27790.3** | 5 | | 27591 | 27669.5 | 30 |
| X-n106-k14 | LKH | 0.9 | 26495 | 26730.1 | 0 | 10 | 26426 | 26557.6 | 0 | 105 | 26381 | 26438.3 | 0 |
| 26362 | NeuroLKH | | 26447 | **26712.8** | 0 | | 26396 | **26528.5** | 0 | | 26381 | **26428.5** | 0 |
| X-n110-k13 | LKH | 0.4 | 14971 | 15216.2 | 2 | 3 | 14971 | **15073.3** | 31 | 29 | 14971 | 15020.4 | 53 |
| 14971 | NeuroLKH | | 14971 | **15207.3** | 2 | | 14971 | 15074.2 | 32 | | 14971 | 15022.3 | 58 |
| X-n115-k10 | LKH | 0.2 | 12750 | 12838.3 | 0 | 2 | 12747 | **12778.3** | 14 | 17 | 12747 | **12770.3** | 46 |
| 12747 | NeuroLKH | | 12747 | **12837.6** | 1 | | 12747 | 12783.9 | 14 | | 12747 | 12771.8 | 40 |
| X-n120-k6 | LKH | 0.3 | 13332 | 13547.4 | 1 | 2 | 13332 | 13394.3 | 10 | 21 | 13332 | 13358.6 | 40 |
| 13332 | NeuroLKH | | 13333 | **13519.9** | 0 | | 13332 | **13389.7** | 5 | | 13332 | **13352.9** | 33 |
| X-n125-k30 | LKH | 3.1 | 56167 | 56690.8 | 0 | 31 | 55733 | 56041.8 | 0 | 335 | 55546 | 55813.0 | 0 |
| 55539 | NeuroLKH | | 56011 | **56624.7** | 0 | | 55645 | **55981.7** | 0 | | 55539 | **55779.7** | 1 |
| X-n129-k18 | LKH | 0.8 | 29173 | 29635.5 | 0 | 8 | 28967 | 29257.5 | 0 | 86 | 28948 | 29108.8 | 0 |
| 28940 | NeuroLKH | | 29160 | **29566.1** | 0 | | 28948 | **29224.3** | 0 | | 28948 | **29081.3** | 0 |
| X-n134-k13 | LKH | 1.1 | 11024 | 11215.7 | 0 | 10 | 10931 | 11048.8 | 0 | 94 | 10916 | 10994.8 | 1 |
| 10916 | NeuroLKH | | 11023 | **11194.9** | 0 | | 10941 | **11044.6** | 0 | | 10916 | **10987.1** | 1 |
| X-n139-k10 | LKH | 0.4 | 13670 | 13894.9 | 0 | 3 | 13605 | 13713.6 | 0 | 33 | 13590 | 13660.4 | 5 |
| 13590 | NeuroLKH | | 13672 | **13871.1** | 0 | | 13605 | **13702.6** | 0 | | 13590 | **13657.0** | 6 |
| X-n143-k7 | LKH | 0.5 | 15765 | **16186.5** | 0 | 5 | 15737 | 15910.4 | 0 | 50 | 15711 | 15812.4 | 0 |
| 15700 | NeuroLKH | | 15781 | 16208.1 | 0 | | 15726 | **15885.9** | 0 | | 15726 | **15787.3** | 0 |
| X-n148-k46 | LKH | 0.9 | 43833 | 44382.4 | 0 | 9 | 43485 | 43819.2 | 0 | 89 | 43448 | 43635.2 | 18 |
| 43448 | NeuroLKH | | 43809 | **44283.0** | 0 | | 43514 | **43818.1** | 0 | | 43448 | **43634.7** | 19 |
| X-n153-k22 | LKH | 1.7 | 21328 | 21559.2 | 0 | 15 | 21236 | 21326.8 | 0 | 156 | 21225 | **21263.6** | 0 |
| 21220 | NeuroLKH | | 21298 | **21493.7** | 0 | | 21240 | **21311.1** | 0 | | 21225 | 21272.1 | 0 |
| X-n157-k13 | LKH | 0.5 | 16903 | 17008.7 | 0 | 4 | 16876 | 16911.0 | 8 | 40 | 16876 | 16893.4 | 40 |
| 16876 | NeuroLKH | | 16900 | **17006.8** | 0 | | 16876 | **16904.9** | 14 | | 16876 | **16889.0** | 52 |
| X-n162-k11 | LKH | 0.3 | 14179 | **14362.6** | 0 | 3 | 14156 | **14225.2** | 0 | 26 | 14138 | **14196.8** | 6 |
| 14138 | NeuroLKH | | 14190 | 14388.8 | 0 | | 14161 | 14245.3 | 0 | | 14138 | 14213.9 | 2 |
| X-n167-k10 | LKH | 0.6 | 20826 | 21319.8 | 0 | 7 | 20583 | 20863.2 | 0 | 65 | 20557 | 20749.5 | 1 |
| 20557 | NeuroLKH | | 20687 | **21270.5** | 0 | | 20592 | **20857.8** | 0 | | 20557 | **20740.3** | 1 |
| X-n172-k51 | LKH | 1.2 | 46141 | 46679.2 | 0 | 11 | 45742 | 46078.0 | 0 | 122 | 45607 | 45840.5 | 5 |
| 45607 | NeuroLKH | | 46134 | **46533.1** | 0 | | 45707 | **45994.7** | 0 | | 45607 | **45783.9** | 3 |
| X-n176-k26 | LKH | 3.6 | 48035 | 48819.7 | 0 | 33 | 47930 | **48273.6** | 0 | 353 | 47840 | **48090.3** | 0 |
| 47812 | NeuroLKH | | 48147 | **48726.3** | 0 | | 47950 | 48279.7 | 0 | | 47812 | 48098.9 | 1 |
| X-n181-k23 | LKH | 0.5 | 25677 | 25829.7 | 0 | 4 | 25611 | 25691.2 | 0 | 42 | 25582 | 25645.3 | 0 |
| 25569 | NeuroLKH | | 25691 | **25822.8** | 0 | | 25603 | **25685.9** | 0 | | 25577 | **25641.2** | 0 |
| X-n186-k15 | LKH | 1.0 | 24297 | **24882.6** | 0 | 10 | 24227 | 24528.3 | 0 | 104 | 24149 | **24359.6** | 0 |
| 24145 | NeuroLKH | | 24511 | 24911.5 | 0 | | 24178 | **24523.0** | 0 | | 24147 | 24361.7 | 0 |
| X-n190-k8 | LKH | 0.9 | 17187 | 17418.0 | 0 | 8 | 17065 | 17275.4 | 0 | 84 | 16993 | 17155.2 | 0 |
| 16980 | NeuroLKH | | 17160 | **17410.0** | 0 | | 17041 | **17259.8** | 0 | | 16985 | **17145.1** | 0 |
| X-n195-k51 | LKH | 1.4 | 44911 | 45594.9 | 0 | 11 | 44437 | 44799.6 | 0 | 117 | 44334 | 44558.1 | 0 |
| 44225 | NeuroLKH | | 44685 | **45244.5** | 0 | | 44422 | **44688.0** | 0 | | 44237 | **44524.8** | 0 |
| X-n200-k36 | LKH | 4.3 | 59329 | 59984.3 | 0 | 39 | 58919 | 59174.2 | 0 | 405 | 58643 | **58927.5** | 0 |
| 58578 | NeuroLKH | | 59229 | **59803.9** | 0 | | 58844 | **59104.6** | 0 | | 58694 | 58937.4 | 0 |

Table S.6: CVRPLIB results (continued)

| Name | Method | LKH with 100 trials as time limit | | | | LKH with 1000 trials as time limit | | | | LKH with 10000 trials as time limit | | | |
|---|---|---|---|---|---|---|---|---|---|---|---|---|---|
| | | Time | Best | Avg | Suc | Time | Best | Avg | Suc | Time | Best | Avg | Suc |
| X-n204-k19 19565 | LKH | 0.6 | 19795 | 20159.5 | 0 | 5 | 19718 | 19880.5 | 0 | 49 | 19662 | 19777.9 | 0 |
| | NeuroLKH | | 19794 | **20076.7** | 0 | | 19692 | **19857.7** | 0 | | 19583 | **19776.3** | 0 |
| X-n209-k16 30656 | LKH | 0.9 | 31259 | 31648.1 | 0 | 9 | 30818 | 31214.9 | 0 | 93 | 30700 | 31028.9 | 0 |
| | NeuroLKH | | 31163 | **31555.8** | 0 | | 30864 | **31140.2** | 0 | | 30722 | **30969.3** | 0 |
| X-n214-k11 10856 | LKH | 2.6 | 11727 | 12131.2 | 0 | 23 | 11147 | **11487.5** | 0 | 229 | 10974 | **11182.9** | 0 |
| | NeuroLKH | | 11702 | **12128.5** | 0 | | 11235 | 11498.2 | 0 | | 10988 | 11214.2 | 0 |
| X-n219-k73 117595 | LKH | 1.4 | 117821 | 118242.7 | 0 | 10 | 117595 | 117790.2 | 1 | 101 | 117595 | 117684.3 | 3 |
| | NeuroLKH | | 117046 | **117998.3** | 0 | | 117622 | **117733.2** | 0 | | 117595 | **117654.8** | 4 |
| X-n223-k34 40437 | LKH | 1.4 | 41250 | 41880.6 | 0 | 12 | 40766 | 41087.1 | 0 | 127 | 40560 | **40818.7** | 0 |
| | NeuroLKH | | 41066 | **41662.6** | 0 | | 40641 | **41022.8** | 0 | | 40563 | 40821.3 | 0 |
| X-n228-k23 25742 | LKH | 1.6 | 26051 | **26541.4** | 0 | 15 | 25863 | 26037.7 | 0 | 150 | 25781 | 25910.4 | 0 |
| | NeuroLKH | | 26067 | 26614.9 | 0 | | 25835 | **26030.5** | 0 | | 25791 | **25907.7** | 0 |
| X-n233-k16 19230 | LKH | 0.5 | 19615 | 19885.4 | 0 | 4 | 19379 | 19599.1 | 0 | 39 | 19305 | 19477.2 | 0 |
| | NeuroLKH | | 19499 | **19831.7** | 0 | | 19381 | **19597.4** | 0 | | 19324 | **19473.2** | 0 |
| X-n237-k14 27042 | LKH | 0.8 | 27381 | 27829.6 | 0 | 7 | 27164 | 27406.5 | 0 | 65 | 27050 | 27276.5 | 0 |
| | NeuroLKH | | 27324 | **27789.5** | 0 | | 27124 | **27402.8** | 0 | | 27042 | **27240.0** | 1 |
| X-n242-k48 82751 | LKH | 2.2 | 84353 | 85218.4 | 0 | 19 | 83419 | 83826.9 | 0 | 198 | 83045 | 83401.3 | 0 |
| | NeuroLKH | | 84090 | **84685.6** | 0 | | 83299 | **83743.7** | 0 | | 83042 | **83357.2** | 0 |
| X-n247-k50 37274 | LKH | 2.8 | 37681 | 38206.5 | 0 | 26 | 37353 | 37701.6 | 0 | 280 | 37289 | 37457.1 | 0 |
| | NeuroLKH | | 37629 | **38118.3** | 0 | | 37326 | **37638.8** | 0 | | 37292 | **37454.3** | 0 |
| X-n251-k28 38684 | LKH | 1.3 | 39394 | 39831.8 | 0 | 11 | 39010 | 39274.9 | 0 | 117 | 38838 | **39067.3** | 0 |
| | NeuroLKH | | 39277 | **39720.2** | 0 | | 38988 | **39259.8** | 0 | | 38887 | 39069.3 | 0 |
| X-n256-k16 18839 | LKH | 2.4 | 19931 | 20953.7 | 0 | 17 | 19150 | 19519.9 | 0 | 148 | 18926 | 19164.9 | 0 |
| | NeuroLKH | | 19681 | **20730.2** | 0 | | 19046 | **19433.9** | 0 | | 18889 | **19143.3** | 0 |
| X-n261-k13 26558 | LKH | 1.2 | 27395 | 27891.3 | 0 | 13 | 26966 | 27367.3 | 0 | 150 | 26686 | 27104.9 | 0 |
| | NeuroLKH | | 27174 | **27746.3** | 0 | | 26749 | **27308.2** | 0 | | 26661 | **27074.4** | 0 |
| X-n266-k58 75478 | LKH | 4.1 | 77457 | 78371.5 | 0 | 35 | 76117 | 76718.3 | 0 | 359 | 75803 | 76193.4 | 0 |
| | NeuroLKH | | 76864 | **77879.7** | 0 | | 76175 | **76582.7** | 0 | | 75876 | **76187.2** | 0 |
| X-n270-k35 35291 | LKH | 1.7 | 35999 | 36580.2 | 0 | 14 | 35513 | 35870.2 | 0 | 142 | 35407 | 35613.1 | 0 |
| | NeuroLKH | | 35808 | **36425.9** | 0 | | 35509 | **35817.5** | 0 | | 35424 | **35598.2** | 0 |
| X-n275-k28 21245 | LKH | 0.7 | 21455 | 21784.3 | 0 | 5 | 21304 | 21524.7 | 0 | 50 | 21245 | **21422.8** | 1 |
| | NeuroLKH | | 21515 | **21715.7** | 0 | | 21320 | **21512.0** | 0 | | 21281 | 21424.8 | 0 |
| X-n280-k17 33503 | LKH | 2.2 | 34230 | 34932.1 | 0 | 22 | 33790 | 34218.8 | 0 | 229 | 33633 | 33943.8 | 0 |
| | NeuroLKH | | 34071 | **34844.4** | 0 | | 33699 | **34178.3** | 0 | | 33632 | **33943.0** | 0 |
| X-n284-k15 20215 | LKH | 0.7 | 20917 | **21194.0** | 0 | 7 | 20580 | 20862.8 | 0 | 76 | 20381 | 20655.2 | 0 |
| | NeuroLKH | | 20903 | 21199.4 | 0 | | 20609 | **20849.8** | 0 | | 20455 | **20639.5** | 0 |
| X-n289-k60 95151 | LKH | 5.9 | 97877 | 99666.8 | 0 | 53 | 96381 | 97129.8 | 0 | 557 | 95687 | 96226.0 | 0 |
| | NeuroLKH | | 97731 | **99084.0** | 0 | | 96163 | **96998.4** | 0 | | 95754 | **96154.4** | 0 |
| X-n294-k50 47161 | LKH | 2.1 | 48490 | 49351.2 | 0 | 17 | 47575 | 48009.5 | 0 | 176 | 47381 | 47644.4 | 0 |
| | NeuroLKH | | 48093 | **48990.2** | 0 | | 47550 | **47914.8** | 0 | | 47354 | **47616.2** | 0 |
| X-n298-k31 34231 | LKH | 1.7 | 35568 | 36543.4 | 0 | 12 | 34732 | 35199.4 | 0 | 121 | 34343 | 34764.9 | 0 |
| | NeuroLKH | | 35380 | **36292.7** | 0 | | 34656 | **35113.6** | 0 | | 34320 | **34763.9** | 0 |

Table S.7: Solomon benchmark results

| Name | Method | LKH with 100 trials as time limit | | | | LKH with 1000 trials as time limit | | | | LKH with 10000 trials as time limit | | | |
|---|---|---|---|---|---|---|---|---|---|---|---|---|---|
| | | Time | Best | Avg | Suc | Time | Best | Avg | Suc | Time | Best | Avg | Suc |
| R201 | LKH | 0.6 | 1252372 | 1275464.5 | 1 | 4 | 1252372 | 1258897.3 | 8 | 35 | 1252372 | 1254027.4 | 21 |
| 1252372 | NeuroLKH | | 1253210 | **1271983.1** | 0 | | 1252372 | **1257114.3** | 5 | | 1252372 | **1253575.6** | 11 |
| R202 | LKH | 4.5 | 1195297 | 1234362.3 | 0 | 33 | 1191698 | 1207016.3 | 19 | 283 | 1191698 | 1197334.8 | 82 |
| 1191698 | NeuroLKH | | 1193776 | **1221507.1** | 0 | | 1191698 | **1204530.3** | 31 | | 1191698 | **1193964.5** | 87 |
| R203 | LKH | 2.1 | 947357 | 964214.4 | 0 | 16 | 941996 | 948987.2 | 0 | 143 | 939504 | 943864.1 | 6 |
| 939504 | NeuroLKH | | 943363 | **957044.9** | 0 | | 941405 | **947506.7** | 0 | | 939504 | **943832.2** | 3 |
| R204 | LKH | 4.9 | 836241 | 879723.0 | 0 | 37 | 829440 | **846041.2** | 0 | 320 | 825510 | 838430.7 | 2 |
| 825510 | NeuroLKH | | 838945 | **875614.2** | 0 | | 825510 | 846814.1 | 1 | | 825510 | **837939.1** | 8 |
| R205 | LKH | 1.4 | 994429 | 1046294.2 | 1 | 11 | 994429 | 1024682.4 | 8 | 95 | 994429 | 1014571.8 | 40 |
| 994429 | NeuroLKH | | 1003685 | **1038416.6** | 0 | | 994429 | **1022870.4** | 6 | | 994429 | **1009598.9** | 45 |
| R206 | LKH | 1.5 | 913333 | 942722.6 | 0 | 12 | 909820 | 926079.9 | 0 | 104 | 906145 | 918597.5 | 19 |
| 906145 | NeuroLKH | | 913333 | **940668.2** | 0 | | 906145 | **925617.6** | 2 | | 906145 | **918009.4** | 24 |
| R207 | LKH | 6.0 | 908532 | 965102.0 | 0 | 51 | 894793 | 929064.0 | 0 | 445 | 890608 | 915756.4 | 1 |
| 890608 | NeuroLKH | | 903583 | **956950.3** | 0 | | 893384 | **924560.7** | 0 | | 890608 | **913160.8** | 5 |
| R208 | LKH | 2.0 | 726817 | 751164.7 | 2 | 15 | 726817 | 736114.7 | 9 | 125 | 726817 | 731161.9 | 16 |
| 726817 | NeuroLKH | | 727258 | **744832.5** | 0 | | 726817 | **733925.3** | 6 | | 726817 | **730790.3** | 9 |
| R209 | LKH | 1.7 | 918711 | 946581.2 | 0 | 14 | 913141 | 927854.0 | 0 | 113 | 909158 | 920110.0 | 7 |
| 909158 | NeuroLKH | | 914609 | **935769.5** | 0 | | 909158 | **922974.0** | 2 | | 909158 | **917506.1** | 9 |
| R210 | LKH | 1.7 | 951624 | 979061.5 | 0 | 13 | 939373 | 959573.6 | 1 | 114 | 939373 | 953584.1 | 20 |
| 939373 | NeuroLKH | | 939373 | **967980.8** | 1 | | 939373 | **955815.6** | 9 | | 939373 | **950722.1** | 39 |
| R211 | LKH | 5.1 | 910853 | 963151.8 | 0 | 44 | 893168 | 926350.7 | 0 | 378 | 890930 | 914120.6 | 2 |
| 890930 | NeuroLKH | | 909830 | **956837.2** | 0 | | 892988 | **923050.2** | 0 | | 890930 | **912125.8** | 2 |