# OpenReview forum: "NeuroLKH: Combining Deep Learning Model with Lin-Kernighan-Helsgaun Heuristic for Solving the Traveling Salesman Problem"
_NeurIPS.cc/2021/Conference — NeurIPS 2021 Poster_

### Official Review · Reviewer_dK7t · 2021-07-12

**Rating:** 6
**Confidence:** 4

**Summary:**

The paper proposes a framework for solving the traveling salesman problem by combining deep learning with the heuristic Lin-Kernighan-Helsgaun (LKH) algorithm. The basic idea is to use a sparse graph network model to transform edge distance and create edge candidate sets for LKH searching trials. The experiment results show that the proposed framework improves the performance of the LKH algorithm for TSP and its variations.

**Limitations And Societal Impact:**

Some comments:

1) It is known that the LKH algorithm is powerful by itself. Given the marginal improvements (objective value) obtained by the proposed framework, I would expect some more intuitive illustration, such as a detailed analysis of an instance, to demonstrate how such framework adds value to boost or improve the original LKH algorithm.

2) The entire framework is based on the LKH algorithm, which is known to be time-consuming. Usually one purpose of adding a deep learning wrap is to speed up the process. Thus, I would expect the authors to also show that the algorithm can achieve the same performance as LKH while significantly reducing the computational time.

**Main Review:**

Originality: The paper uses the sparse graph network model to learn the edge scores with supervised learning and node penalties with unsupervised learning respectively. The output is then used to guide the original LKH algorithm. The entire framework combines the strength of deep learning and the traditional heuristic approach.

Quality: The paper is complete.

Clarity: The paper is clear to understand and easy to follow.

Significance: The authors claim that the proposed algorithm significantly outperforms the LKH algorithm. However, the experiment results only indicate a less than 0.1% improvement with 1000 instances under the same time limit for TSP when |V| = 100, 200, 500 respectively. Such marginal difference should not be considered statistically significant. This is a major weakness of the paper.


**Time Spent Reviewing:**

10

---

> ### Author Response · Authors · 2021-08-10
> **Response**
>
> We would like to thank you for your time and efforts in reviewing our paper and the valuable comments. We hope our response below could clarify the concerns.
> 1. Regarding significance:
> (1). For small-size TSP with 100, 200 and 500 nodes, improvement more than 0.1% over LKH is not possible for any algorithms since the optimal gap of LKH is already less than 0.1% even with the shortest time limits, as shown in Table 1. Therefore, in such cases, the improvement of objective value is not very meaningful as a measure. Instead, researchers use the relative improvement of the optimal gap as a better indicator. Compared to the original LKH algorithm, NeuroLKH reduces the optimality gaps by at least an order of magnitude in most of our experiments, which is a significant improvement, similar to reducing the error rate by about 10 times for a classification problem.
> (2). More importantly, results of small-size TSP are only a small part of our experiments. For large-size TSP (1000, 2000 and 5000 nodes) with short time limits, the improvements of objective values over LKH are more than or close to 0.1%. On more complicated problems such as CVRP, PDP and CVRPTW, the improvements of objective values are much more significant. We also show the generalization of NeuroLKH to traditional benchmarks.
> (3). In addition, we show that the improvements are statistically significant in the Supplementary Material. The improvements of NeuroLKH over LKH on various problems with different sizes and with different time limits are almost all statistically significant with more than 99% confidence. Please refer to the details in the Supplementary Material (lines 26-30, 60-67).
> 2. Regarding comment 1:
> (1). While showing promising results in our paper and related works, a major limitation of deep neural networks is the black-box nature, meaning that they are hard to explain and interpret with the large number of learnable parameters. Nevertheless, considering the interpretable models for VRPs is an interesting future direction.
> (2). We will add some statistics analysis and case studies into our paper to give some insights about why NeuroLKH improves LKH, such as the average ranking of the optimal edges. For example, regarding TSP with 100 nodes, the average ranking of optimal edges in the original LKH is 1.7. The ideal average ranking would be 1.5 since the two optimal edges for each node would be the first and the second in the ranks. NeuroLKH reduces the average ranking to 1.55, which explains its good performance.
>
> 3. Regarding comment 2:
> For Combinatorial Optimization Problems, there is usually a trade-off between performance and computational time. In Tables 1, 2 and 3, we use the same computational time to show the performance improvement of NeuroLKH. In Figures S.1 and S.2 of the Supplementary Material, we plot the performance of NeuroLKH and the baseline algorithms for solving various routing problems with different numbers of nodes against different running time to visualize the improvement process. As shown in the plots, with the same performance (i.e. objective value), NeuroLKH reduces the computational time significantly. For example, for the results of LKH on TSP 100 in Table 1 (with 1, 10, 100 and 1000 trials), NeuroLKH only uses 37.4%, 28.9%, 17.0% and 32.5% of the run time to achieve the same or better performance, respectively (9.4%, 13.7%, 23.6%, 39.9% for TSP5000). For all the settings of LKH in Tables 1 and 2, NeuroLKH reduces the run time of LKH by 60.1%-90.6% to achieve the same or better performance, which is a very large margin. Please refer to the details in the Supplementary Material (lines 2-15, 57-59). We will move the plots to the main paper with more discussions.

---

> > ### Comment · Reviewer_dK7t · 2021-08-15
> > **Response**
> >
> > Thank you very much for you response. I will raise my rating from 4 to 6.
> >
> > My main concern is that such improvement is not scientifically or practically significant, especially for TSP. However, the results in the Appendix about CVRP, CVRPTW, and CVRPLib are more significant and convincing. I would suggest authors emphasizing those results and moving them to the main paper.

---

> > > ### Author Response · Authors · 2021-08-16
> > > **Response**
> > >
> > > Thank you very much for your quick response and acknowledging the significance of our results. We will definitely follow your suggestion and move the results to the main paper with more discussions.

---

### Official Review · Reviewer_rptX · 2021-07-15

**Rating:** 7
**Confidence:** 5

**Summary:**

This paper proposes a machine learning approach to enhance the performance of a traditional heuristic LKH to solve routing problems. Specifically, a Sparse Graph Network is trained to select candidate edges for LKH to refine solutions. The experiments confirm that the proposed method can improve the performance of the original LKH and the VSR-LKH (based on RL) in general.

**Limitations And Societal Impact:**

The authors have adequately addressed the limitations of the proposed method, especially the generalization of the method to other TSP instances with different characteristics and other routing problems.

**Main Review:**

This paper combines the merits of both ML and traditional optimization techniques to solve large and hard routing problems. The approach sounds reasonable and appealing. The proposed method is well described, and the experiment is extensive.

1. For the Sparse Graph Network to work, the original TSP instances need to be pre-processed, i.e., only keeping \gamma shortest edges for each node and removing others. I guess the parameter \gamma has a large impact on the performance of the model. Why \gamma is set to 20 in the experiments, and what is the impact if changing \gamma to a different value?

2. Following the previous comment, sparsifying TSP graphs based on edge distance might lead to a sub-optimal solution in the first place, as mentioned by the authors that a certain amount of edges in the optimal tour were removed in pre-processing. Techniques [1,2] that sparsify TSP graphs based on machine learning might work better here.

    [1] Sun, Yuan, et al. "Generalization of machine learning for problem reduction: a case study on travelling salesman problems." OR Spectrum (2020): 1-27.

    [2] Fitzpatrick, James, Deepak Ajwani, and Paula Carroll. "Learning to Sparsify Travelling Salesman Problem Instances." International Conference on Integration of Constraint Programming, Artificial Intelligence, and Operations Research. Springer, Cham, 2021.

3. In Equation (6), why the parameter C is set to 10? Justification is required.

4. In Tables 1 and 2, the improvement in terms of solution gaps is quite small, partially because all the methods can find very near-optimal solutions. I think some results from Appendix A should be presented and discussed in the main paper to justify this, for example the converging curves and detailed statistical test results.

5. The comparison between the LKH variants ignoring preprocessing time in Appendix A is interesting and important, which demonstrates that the improvement of the proposed method comes from a better ML prediction rather than simply running for more trials. I think this result should be discussed in the main paper in particular.

6. In Figure S.1, why does NeuroLKH start from with an objective value much smaller than that of other methods? From my understanding, all the methods start with the same random generated solution, and so the initial objective value should be the same.

7. In figure S.1, why do LKH and VSR-LKH ignoring pre-processing time terminate earlier? What happens if letting them running for the same amount of time?

**Time Spent Reviewing:**

2.5

---

> ### Author Response · Authors · 2021-08-10
> **Response**
>
> We would like to thank you for your time and efforts in reviewing our paper and the valuable comments. We hope our response below could clarify the concerns.
> 1. For TSP, we choose $\gamma=20$ to include most of the edges in the optimal tours into the sparse graph, which results in only 0.01% of the optimal edges missed in the sparse graph for the training dataset, as stated in the paper (lines 227-229). In our TSP experiments with different $\gamma$, $\gamma$ larger than 20 only improves the performance marginally but increases the computational time. And for other routing problems, we find similar results therefore we use $\gamma=20$ for consistency. We find that the network can hardly give a high edge score to an edge with very large Euclidean distance and include it into the candidate set. Therefore larger $\gamma$ is not needed and $\gamma$ does not have a large impact on the performance as long as it is not too small (e.g. less than 20). We will add the results for different $\gamma$ in the Supplementary Material.
> 2. Yes, combining with advanced sparsifying techniques will further improve the performances. This is a very interesting future direction and we will include this point and references into the paper as future work.
> In the paper, we keep the sparsifying process simple and applicable to other routing problems by using the shortest edges in Euclidean distances.
> 3. In the original LKH algorithm, a subgradient optimization process is used to optimize the node penalties iteratively until convergence for each instance. In this process, we find that the penalties are usually between -10 and 10 for the instances in our training dataset (the coordinates are always between 0 and 1). Therefore we use C=10 in our experiments. And while testing for instances with different coordinate scales, we scale the instances to make the coordinates between 0 and 1. The aspect ratio is fixed so that the objective value is just scaled by a constant. We tried larger C which did not affect the performance much. We will add the results for different C in the Supplementary Material.
> 4. Yes, we will add these results and discussions in the main paper.
> 5. Yes, we will add these results and discussions in the main paper.
> 6. For LKH, VSR-LKH and NeuroLKH algorithms, in each trial, one initial solution is randomly generated and improved on. In Figure S.1, the data points are the resulting objective value after each trial, not the improving process within each trial. Therefore, the first point of each curve is the objective value returned by the corresponding algorithm after the first trial, not the initial solution. We will clarify this in the paper.
> 7. We use the same time limits to run LKH, VSR-LKH and NeuroLKH and plot the results. And for the ignoring pre-processing time plots, we use the same results to plot with the run time as the total time minus the pre-processing time. NeuroLKH has a much shorter pre-processing time. Therefore it seems that LKH and VSR-LKH ignoring pre-processing time terminate earlier (the actual run time including the pre-processing time is the same for the three algorithms). Letting them run for the same amount of time will show a similar trend (NeuroLKH still outperforms both VSR-LKH and LKH by similar margins). And we will change it to the same time limit in the paper.

---

> > ### Comment · Reviewer_rptX · 2021-08-20
> > **Response**
> >
> > Thanks for the clarification and further investigation on parameter sensitivity. I am convinced that this is a nice paper.

---

> > > ### Author Response · Authors · 2021-08-20
> > > **Response**
> > >
> > > Thank you very much for acknowledging the value of our paper. We will follow your suggestion to revise our paper.

---

### Official Review · Reviewer_U8cn · 2021-07-16

**Rating:** 7
**Confidence:** 4

**Summary:**

This paper proposes a neural version of the classic LKH algorithm, where the subgradient optimization of LKH is replaced by a sparse GNN that predicts edge candidates and node penalties that can directly be used by the LKH search algorithm. The edge candidate predictions are learned supervised from optimal solutions whereas the penalties are learned 'unsupervised' to push the node degrees of the 1-tree induced by the penalties towards 2 (I consider this is more like RL where the deviation from node degree 2 determines the reward). After training, the NeuroLKH algorithm gives better results than the standard LKH algorithm given equal time, for the travelling salesman problem (TSP) and different vehicle routing problems (VRPs).

As the node penalties are only used for TSP, the major contribution seems to show the empirical benefit of the edge candidate predictions. While this is not a new idea in itself, the combination with LKH is and as the empirical results are very promising, I think this paper makes valuable progress and therefore should be accepted.

**Limitations And Societal Impact:**

The major limitation is the dependence on the actual distribution of the instances and is discussed in the discussion. While not discussed, I do not see any direct negative societal impact.

**Main Review:**

**Originality**
On one hand, this paper presents a nice and simple idea of taking the subgradient objective for LKH but amortize its optimization over a set of instances to predict the node penalties for new instances without needing the subgradient optimization. On the other hand, the idea of learning to predict an edge candidate set is closely related to the cited works [19,21] and the differences are merely in architecture (most notably using a sparse network for increased scalability), so this in itself is not very novel. The combination with LKH, while relatively straightforward, has not been done before as far as I know.

**Quality**
The paper is technically of high quality. The claims are supported by a large number of experiments and ablations on different datasets, instance distributions and problem sizes. Care has been taken to present fair comparison by forcing running times equal, and helpful ablations are included in the appendix. My major concern is that the learned node penalties are only used with TSP, for which the practical differences in performance seem small. For CVRP and TSPTW, the difference with LKH and other baselines is much more convincing, but this seems only result of the edge selection by the network. Therefore I consider the main contribution to show the strong empirical performance of using a carefully implemented neural network for edge selection in combination with an existing heuristic (LKH).

**Clarity**
Overall I found this paper very clear. Whereas a lot of space is used for the extensive empirical evaluation, I find most of the important details very clear, without going more in depth than necessary. The background on the LKH algorithm is very helpful and Figure 1 is very clear in pointing out the differences and similarities between LKH and NeuroLKH. I do have some questions about the unsupervised training and I think some of the mathematical notation can be improved (see below).

**Significance**
This paper shows promising results on relatively large instances for different problems using a relatively simple methodology, namely the selection of edges by a graph neural network (and the prediction of node penalties used to change the edge cost for TSP), which can then be used by LKH, a relatively general solver for VRP problems. Whereas the ideas are quite simple, I consider it valuable to demonstrate the empirical performance of this methodology, which encourages further exploration on larger or other problems, as well as combination with other, more powerful solvers than LKH (which do exist and could/should be mentioned in the paper, see e.g. Hybrid Genetic Search for CVRP, Vidal 2020).

**Minor comments**
- I find the equations (1), (2) a bit confusing. I think attn_ij^l is a vector which may be helpful to mention explicitly (and as such exp(.) is also element wise). Also, it may be worth to mention the dimension over which BatchNorm operates / that it is not an element wise function as in the equations.
- Also the notation (i,m) in E* is confusing since it looks as if i is a summation index while it is actually fixed. I suggest introducing N(i) as neighbours of i and sum over m in N(i).
- I think some more explanation could be helpful for the loss in eq. (8). I understand now that it is correct but I was a bit surprised at first not to see absolute signs or a squared difference (as in a MSE loss).
- L213, it would be helpful to explain/remind why the candidate sets are of much better quality than LKH (because they are trained on 'optimal' solutions, which I missed the first time reading).

**Questions**
- Why do you consider the attention 'cheap' (L160)? I think it is actually more expensive as it computes a softmax per individual feature.
- How does the unsupervised learning of the penalties work exactly? Does the model predict penalties, then find a minimum 1-tree, then use this for the loss? How are min-1-trees found for a batch of instances? What is the batch size used? This sounds very costly. Also, I would consider this more like a reinforcement learning problem then where the action is deciding the penalties.
- Do I understand correctly that starting from a sparse graph, some edges can never be in a solution? How does this affect solution quality? Especially for large scale VRPTW, I can imagine the optimal solution would not use the gamma = 20 shortest outgoing edges?
- How does C (eq. 6) relate to the scale of the coordinates / size of the problem instance? This seems important as the penalty and thus C is actually in distance units.
- Why can't node penalties be used with CVRP? When you fix the penalty for the depot to 0 (as number of visits to the depot may vary) I do not immediately see a problem?
- The penalties affect the quality of the alpha measure, but do I understand correctly that this is not used in NeuroLKH as the edge set is predicted directly? So the penalties are only used to change the edge distances used in the search?

**Time Spent Reviewing:**

5

---

> ### Author Response · Authors · 2021-08-10
> **Response**
>
> We would like to thank you for your time and efforts in reviewing our paper and the valuable comments. And we will follow the suggestions in the minor comments to revise our paper and add the mentioned references. We hope our response below could clarify the concerns in the questions.
> 1. By “cheap version of attention”, we mean that point-wise attention does not need to calculate the attention weights with the dot product of query and key vectors. The softmax function is calculated over the $\gamma=20$ dimension which is a small constant. However, it is only cheap when the same amount of attention weights is required (compared to $D$-head multi-head attention, where $D$ is the hidden dimension). We will rephrase this part to be more accurate.
> 2. (1). Yes, the deep model infers the node penalties for each instance in a batch. Then we find the minimum 1-tree for each instance graph with the node penalties. For training the SGN network, we use a batch size of 4000/|V| (|V| is the number of nodes in the TSP instance), which allows us to use one RTX-2080Ti GPU. To get the minimum 1-tree, we do it sequentially for each instance in the batch, which is a super fast process using Prim’s algorithm  ($O(|V|^2)$ time complexity, less than 0.01s for one TSP instance with 500 nodes). In the original LKH algorithm, it keeps finding the minimum 1-tree and optimizes the node penalties until convergence for each instance.
> (2). We agree with you that the setting is similar to reinforcement learning if we consider inferring the node penalties as sampling an action and the average of $-|d_i(\pi) - 2|$ as the reward (we will discuss the reward part in the paper following the third point in the minor comments). We consider it as unsupervised learning in our paper as it does not fit into typical reinforcement learning algorithms such as policy gradient and Q learning. It is a meaningful insight and we will add more discussion in the paper.
> 3. (1). Some edges can never be in the edge candidate set but any edges can be in the solution. Because in each LKH search trial, one initial solution is constructed where the edges in this tour do not need to be in the edge candidate set. Therefore any solution could be found in the multiple LKH search trials.
> (2). For TSP, we choose $\gamma=20$ to include most of the edges in the optimal tours into the sparse graph, which results in only 0.01% of the optimal edges missed in the sparse graph for the training dataset, as stated in the paper (lines 227-229). In our TSP experiments with different $\gamma$ , $\gamma$ larger than 20 only improves the performance marginally but increases the computational time. And for other routing problems, we find similar results therefore we use $\gamma=20$ for consistency. We find that the network can hardly give a high edge score to an edge with very large Euclidean distance and include it into the candidate set. Therefore larger $\gamma$ is not needed. We will add the results for different $\gamma$ and some discussion in the Supplementary Material.
> 4. In the training set, the coordinates are always between 0 and 1. And while testing for instances with different coordinate scales (such as for TSPLIB), we scale the instances to make the coordinates between 0 and 1. The aspect ratio is fixed so that the objective value is just scaled by a constant.
> In the original LKH algorithm, a subgradient optimization process is used to optimize the node penalties iteratively until convergence for each instance. In this process, we find that the penalties are usually between -10 and 10 (for different sizes). Therefore we use C=10 in our experiments. We tried larger C which did not affect the performance much. We will add the results for different C and some discussions in the Supplementary Material.
> 5. The application of node penalties is based on the fact that the minimum 1-tree with all node degrees equal to 2 is an optimal tour for the TSP instance. For CVRP and other routing problems, this is no longer true due to the capacity and other constraints, and the minimum 1-tree could be infeasible as a solution.
> 6. Yes, in the original LKH algorithm, the node penalties affect the quality of the alpha measure (the edge candidate set) and the change of edge distances in the search. In NeuroLKH, the penalties are only used to change the edge distances used in the search. In [13], the author finds that the quality of solutions is improved by the changed edge distances. We find similar results in our experiments.

---

### Official Review · Reviewer_2rWn · 2021-07-16

**Rating:** 6
**Confidence:** 3

**Summary:**

The current paper proposes a new algorithm that combines deep learning with the heuristic solver for solving various vehicle routing problems, TSP, CVRP, PDP, CVRPTW. The method proposes a Sparse Graph Network (SGN) that produces edge scores and node penalties, which are then used to the edge candidate set and transformed edge distance. Once the edge candidate set and the transformed edge distances are computed, a conventional /lambda-opt algorithm is repeatedly used to produce the final best routing solution. The proposed sore functions are trained with small-sized problems sampled uniformly using supervised learning and unsupervised learning. Finally, the trained model is used to solve the test problems whose node distribution is different and whose node number is large. The proposed algorithm consistently outperforms its baseline, LKH, and VSR-LKH.

**Limitations And Societal Impact:**

The author comments that the proposed method generalizes well to large sizes and different distributions of nodes and does not generalize well to other distributions of demands and time windows without further training.

To achieve the generalization capability of the trained model over other aspects, the specially designed architecture or specially designed training scheme are generally required. The current study does not explicitly mentions such additional efforts but evaluates their performance. It would be good if the authors discuss the indented design choice to achieve what desirable performance and validate that argument through several ablation studies.


**Main Review:**

<Originality>
Combining a well-known and effective search heuristic with a learning-based score model sounds reasonable and effective in solving the target problem. In solving vehicle routing problems, various approaches devise improvement heuristics. While comparing with these improvement heuristics, please mention the unique novelty of the proposed method.

<Quality>
The architecture of SGN is pretty simple. It would be good author justify the design choice of the architecture and the hyperparameters used.

Q1. What is the rationale behind choosing the proper number of $\gamma$?

Q2. Why embedding layer uses skip connection?

Q3. How to determine the number of layers? When the size of the graph changes, the same number of layers is used? The number of layers can be possibly jointly determined with the sparsity parameter $\gamma$?

The SGN is trained with both supervised loss and unsupervised loss. I have several concerns about this approach:

Q4. The biggest concern is that the method requires the optimal solution or a good solution to produce a good supervise label. What are the benefits of using the imitation learning approach comparing to reinforcement learning approach? It would be nice to compare the pros and cons and justify your selection.

Q5. What is the rationale behind using the unsupervised loss? How to compute $d_i(\pi)$ and is ti differentiable? And why this factor should close to 2?

Q6. Can you train both edge score function and node penalty function simultaneously using reinforcement learning without accessing the optimum solution? What is more beneficial for generalizing over different node sizes and node distribution between imitation learning and reinforcement learning?

<Clarity>
The paper is well written and well organized.

<Significance >
The current paper replaces the existing empirical procedure for constructing edge candidate sets with a neural network-based edge candidate selection procedure. The proposed sparse graph network is a simple variant of a graph neural network with limited edge connectivity. I believe replacing a rule-based heuristic module with a learning-based module can naturally result in performance improvement. The paper does not propose a different way of problem-solving but proposes a straightforward extension to the original approach; the technical significance and novelty are limited.

In addition, the proposed method does not effectively generalize over the new test problems with different node distribution and different node numbers. The proposed method seems to work better than other baselines but often fails to solve the problems. In addition, it is hard to see that the method is completely transferable in that it requires fine-tuning.


**Time Spent Reviewing:**

8 hours

---

> ### Author Response · Authors · 2021-08-10
> **Response (2/2)**
>
> 10. Regarding fine-tuning:
> (1). As shown in our TSP experiments, the edge scores generalize well to much larger graphs and different node distributions directly without any fine-tuning. The node penalties can be viewed as an additional technique only applied to problems with specific distributions and number of nodes, which is the goal for most existing deep learning works. And the learned node penalties in one model actually work well for a wide range of graph sizes (100-500 nodes) directly without fine-tuning. Furthermore, it only requires *less than one minute* to fine-tune a very small amount of parameters without supervision for the problems with one size (a class of instances instead of one instance), even for large TSP with 5000 nodes, as shown in lines 264-270.
> (2). Without the learned node penalties, NeuroLKH still outperforms LKH on different numbers of nodes and different node distributions (refer to the experiments for TSPLIB in lines 290-311, the same model and learned parameters are used for different sizes and different node distributions without any fine-tuning, this testing dataset contains instances with graph sizes much larger than the training sizes and various node distributions not used in training). And for the settings in Tables 1 and 2, NeuroLKH with only the learned edge scores still outperforms the original LKH by large margins consistently (also one model without any fine-tuning for different sizes). We will add these results to the Supplementary Material.
> (3). In addition, the node penalties are only applied to TSP instead of other routing problems, therefore the results in Table 3 also show the direct generalization of NeuroLKH to much larger sizes without fine-tuning.
> 11. Regarding generalization:
> (1). We would like to emphasize that it is hard for all learning based methods to generalize to distributions different from training. For example, as reported in [31], the AM model [22] which performs very well on random instances generalizes poorly to instances in TSPLIB. In contrast, while training on random instances, our NeuroLKH still shows reasonably good performance on these instances very different from training ones.
> (2). To the best of our knowledge, generalization directly to very different distributions of demands and time windows is not achieved in any existing deep learning works. Most other deep learning works only consider demands uniformly generated from integers {1..9} for CVRP. As far as we know, the only two other deep learning works [17, 21] considering different distributions of demands (as the dataset in [29]) are both trained on the same distributions as the testing ones and not directly generalized (plus they both train the model for one fixed graph size, in contrast, one NeuroLKH model performs well on a wide range of graph sizes).
> (3). And generalization directly to much larger sizes without much performance degradation is hardly achieved in other deep learning works. Note that most existing works train different models for problems with different sizes and most of them are only tested on relatively small problems (with the exception of [9] which requires complicated design and the performances deteriorate rapidly with the increase of graph sizes). In contrast, our NeuroLKH generalizes well to much larger graph sizes.
> 12. Regarding limitations:
> We purposely avoid making the architecture unnecessarily complicated, which does not mean it is not well designed. With the very effective architecture of SGN, we show the performance better than the strong heuristic LKH, the good generalization to much larger sizes and different node distributions. We will follow the suggestion and add more intuitive explanations of our design, results and discussions for the alternative architectures and hyper-parameters in our experiments to the Supplementary Material.

---

> > ### Comment · Reviewer_2rWn · 2021-08-27
> > **Thank you for the detailed response**
> >
> > The authors have addressed most of my concerns. I raised my score from 5 to 6. However, it is still not clear how the proposed scheme is able to generalize to large-scale problems. As the authors mentioned, most deep learning does not generalize well when the data distribution changes. The Paper does not fully explain why the proposed method can generalize well to large-sized problems. For example, it can be related to (for example) the specific architecture having the size invariance or order transferability, etc.

---

> > > ### Author Response · Authors · 2021-08-29
> > > **Response**
> > >
> > > Thank you very much for the response. We are very glad that most of your concerns have been addressed. Regarding the last concern on the reasons behind the good generalization ability of our proposed method, like we briefly discussed in the point 8.(2) of our response, it is more related to the task formulation and the architecture design. We find that learning in a non-sequential way with a sparse graph is crucial, where all edge scores are predicted at once. This could be explained by the following two reasons. (1) In a sequential way, the decision (such as visiting which node or modifying which part of a solution) for each state is made sequentially, where the new state depends on the previous decision. If a state encountered during testing (and states similar to this one) is not encountered during training, which is very common in generalization, its subsequent actions and the resulting solution could suffer from the impact of the compounding error and could be much worse. In contrast, with the non-sequential way, the edge scores for all nodes are predicted together at once in our method. Therefore, the misprediction of the edges for one node will not affect those of other nodes. (2) In the sparse graph of our method, the number of directed edges pointed from each node is always the same for different graph sizes. Therefore, when the graph size changes, the global structure is different but the local structure remains the same, even for much larger graph sizes not used during training. Due to this local invariance, with the specifically designed Sparse Graph Network, the patterns learned on small-size instances effectively generalize to large-size instances. We will add more discussions on this point in our paper.

---

> ### Author Response · Authors · 2021-08-10
> **Response  (1/2)**
>
> We would like to thank you for your time and efforts in reviewing our paper and the valuable comments. We hope our response below could clarify the concerns.
>
> 1. Regarding originality:
> Most existing deep learning methods for improvement heuristics (e.g. [5, 10, 31]) focus on learning which action to take at each improvement step, which is computationally costly and hard to learn. In contrast, we aim at improving the strong traditional solver LKH by providing crucial initial information to guide the original LKH search process.  The inference of our NeuroLKH is very efficient since the edge scores and node penalties are inferred only once for each instance. NeuroLKH outperforms LKH, which cannot be achieved by most existing learning based improvement methods, and generalizes well to much larger sizes and different node distributions from public benchmarks such as TSPLIB. NeuroLKH is very general and can be applied to many routing problems, such as TSP, CVRP, PDP and CVRPTW as shown in our experiments. We will follow the suggestion to improve our paper, and add more discussion on the unique novelty of our method in the related work section.
> 2. Regarding quality:
> (1). We intentionally avoid making the architecture unnecessarily complicated and hard to understand. Instead, we design the Sparse Graph Network, which is very effective for our task with the sparse directed graphs, and delivers very good performance. Moreover, this design enables good generalization to much larger sizes and different distributions. And we discuss some intuition behind the design of architecture in the paper (such as the design for the opposite-direction counterpart in lines 161-164). We will add more intuitive explanations of our design in the paper.
> (2). In our experiments, we tried other fancier architectures (e.g. with different variants of multi-head attention) and different hyper-parameters. But they do not provide better performances therefore we did not include them in the paper to save space for other more important results. We will follow the suggestion and add some results and discussions for the alternative architectures and hyper-parameters in our experiments to the Supplementary Material.
> 3. Answer for Q1:
> For TSP, we choose $\gamma=20$ to include most of the edges in the optimal tours into the sparse graph, which results in only 0.01% of the optimal edges missed in the sparse graph for the training dataset, as stated in the paper (lines 227-229). In our TSP experiments with different $\gamma$, $\gamma$ larger than 20 only improves the performance marginally but increases the computational time. And for other routing problems, we find similar results therefore we use $\gamma=20$ for consistency. We find that the network can hardly give a high edge score to an edge with very large Euclidean distance and include it into the candidate set. Therefore larger $\gamma$ is not needed. We will add the results for different $\gamma$ and some discussions in the Supplementary Material.
> 4. Answer for Q2:
> The skip connection layers are ubiquitous in deep learning models, which help to mitigate the gradient vanishment and enable the training of deeper networks.
> 5. Answer for Q3:
> We empirically choose the number of layers to be 30, the same as existing works [19, 21], which also works well in our experiments. We use the same number of layers when generalizing across graphs of different sizes. Like other hyper-parameters in deep learning models (such as learning rate, hidden dimension), the number of layers and $\gamma$ are determined based on our experiments and/or the values used in the related works. Same as most other papers, we did not include these details to save space for other more important results.
> 6. Answer for Q4:
> In the existing works of using deep (reinforcement) learning to solve routing problems, reinforcement learning often works well on less studied problems as they can learn heuristics that are better than simple hand-designed ones, less optimized algorithms, and general purpose solvers (on large problems). However, only trained by the rewards from the environment, it is hard to find solutions close to optimal because of the combinatorial nature (the number of tours is exponentially large). As a result, the performance of these models (e.g. [2, 22, 26]) is good but usually far worse than highly optimized algorithms such as LKH. For well-studied problems such as TSP, highly optimized algorithms are available therefore we utilize their solutions to train the deep model, which allows us to outperform these algorithms in turn. In addition, only training for the edge scores in NeuroLKH requires supervision. As shown in the experiments, the learned edge scores generalize well to much larger problems and different node distributions directly without any fine-tuning. Therefore, we only need to train one model with labels for relatively small-size problems, which is relatively easy to obtain. We will follow the suggestion and add more discussions to our paper.
> 7. Answer for Q5:
> The unsupervised training process for node penalties is designed based on the subgradient optimization method for TSP proposed in [12]. $d_i(\pi)$ is the node degrees in the Minimum 1-Tree of the TSP graph induced with the node penalties $\pi$. We use Prim’s algorithm ($O(|V|^2)$ time complexity, |V| is the graph size) to get the Minimum 1-Tree and compute the node degrees in it, which is a super fast process. The node degree $d_i(\pi)$ is not differentiable. Similar to the reward in reinforcement learning, $d_i(\pi)$ does not need to be differentiable. Only the node penalties need to be differentiable. By definition, the Minimum 1-Tree with all node degrees ($d_i$) equal to 2 is an optimal tour for the TSP instance. And the edge distances in the Minimum 1-Tree with node degrees closer to 2 help the searching process to find better solutions, as shown in [13] (The node penalties change the edge distances but not the optimal solution). Therefore we optimize the network to penalize the nodes to make $d_i(\pi)$ closer to 2. Please refer to the details in lines 111-123 (background about subgradient optimization) and lines 177-181 (how we adapt this method to unsupervised training for deep learning models).
> 8. Answer for Q6:
> (1). Like we discussed in Q4, for deep networks learning to solve routing problems, reinforcement learning can train the models to find good solutions. But the performance for large-size problems is usually far from optimal due to the exponentially large solution space to explore. To outperform highly optimized algorithms such as LKH, we need supervised learning to get optimal or near-optimal feedback.
> It would be possible to train the edge scores and node penalties simultaneously without labels. However, in our previous attempts to use reinforcement learning to train edge scores, a large performance drop is hard to avoid due to the reason we discussed above. Therefore we believe that reinforcement learning is better to solve less well-studied problems where optimal and near-optimal labels are unavailable. And the models can find better or comparable solutions to the relatively weak baseline algorithms.
> (2). The generalization ability is more related to the task formulation and the architecture design. We find that learning in a non-sequential way with a sparse graph is crucial, where all edge scores are predicted at once. In contrast, reinforcement learning focuses on sequential decision making, which could suffer from the impact of compounding error. If a state is not encountered before, which is very common in generalization, its subsequent actions and the resulting solution could be much worse.
> 9. Regarding significance:
> (1). We agree with you that it is natural to expect performance gain when a hand-crafted module is replaced by a learned one. In fact, this is the basic motivation of many existing works on deep learning based methods for VRP, where different components such as node selection rule in construction heuristics [7, 22], neighbor selection rule in local search [31] and solution repair procedure in LNS [17] are replaced by learned ones. However, we need to emphasize that it is not trivial and straightforward to identify what to learn and how to learn it, such that good performance can be achieved. In our case, we managed to identify a critical procedure in LKH, i.e. the generation of edge candidate sets, that could be replaced by a learning based module. We carefully designed a neural architecture and the hybrid training scheme (supervised+unsupervised), which outperforms LKH on a wide range of routing problems.
> (2). We don’t understand “The proposed method seems to work better than other baselines but often fails to solve the problems.” If it means that NeuroLKH fails to solve some instances optimally, routing problems are NP-hard and it is very hard to solve all instances optimally in polynomial time. The strong heuristic LKH cannot even give optimality guarantee.
> (3). And as we show below (in the second half of our response), different from most existing deep learning works for routing problems which generalize poorly, our proposed NeuroLKH generalizes reasonably well to much larger graph sizes and different node distributions with better performance than the original LKH algorithm.

---

### Decision · Program_Chairs · 2021-09-27

**Decision:**

Accept (Poster)

**Comment:**

The reviewers all agreed that the combination of learning techniques to augment traditional problem solvers is an exciting direction, and this paper adds to that direction.  Reviewer dK7t was ultimately convinced by the author's response and material in the appendix (Figure S.1) that the contribution was worthwhile even if the objective improvement was small in many of the TSPs.  I think the paper could be improved by directly addressing this in the paper: both the discussion of relative improvement and highlighting the larger improvements in other domains.